# Meiotic prophase length modulates Tel1-dependent DNA double-strand break interference

**Luz María López Ruiz** *, **Dominic Johnson** , **William H. Gittens**, **George G. B. Brown** , **Rachal M. Allison** , **Matthew J. Neale** *

Genome Damage and Stability Centre, School of Life Sciences, University of Sussex, Brighton, United Kingdom

* lm.lopezruiz.llr@gmail.com (LLR); m.neale@sussex.ac.uk (MJN)

**Data Availability Statement:** Processed hotspot average table files and analysis scripts are available at: https://github.com/Neale-Lab/Ndt80_LLR Raw

## Abstract

During meiosis, genetic recombination is initiated by the formation of many DNA double-strand breaks (DSBs) catalysed by the evolutionarily conserved topoisomerase-like enzyme, Spo11, in preferred genomic sites known as hotspots. DSB formation activates the Tel1/ATM DNA damage responsive (DDR) kinase, locally inhibiting Spo11 activity in adjacent hotspots via a process known as DSB interference. Intriguingly, in *S. cerevisiae*, over short genomic distances (<15 kb), Spo11 activity displays characteristics of concerted activity or clustering, wherein the frequency of DSB formation in adjacent hotspots is greater than expected by chance. We have proposed that clustering is caused by a limited number of sub-chromosomal domains becoming primed for DSB formation. Here, we provide evidence that DSB clustering is abolished when meiotic prophase timing is extended via deletion of the *NDT80* transcription factor. We propose that extension of meiotic prophase enables most cells, and therefore most chromosomal domains within them, to reach an equilibrium state of similar Spo11-DSB potential, reducing the impact that priming has on estimates of coincident DSB formation. Consistent with this view, when Tel1 is absent but Ndt80 is present and thus cells are able to rapidly exit meiotic prophase, genome-wide maps of Spo11-DSB formation are skewed towards pericentromeric regions and regions that load pro-DSB factors early—revealing regions of preferential priming—but this effect is abolished when *NDT80* is deleted. Our work highlights how the stochastic nature of Spo11-DSB formation in individual cells within the limited temporal window of meiotic prophase can cause localised DSB clustering—a phenomenon that is exacerbated in *tel1Δ* cells due to the dual roles that Tel1 has in DSB interference and meiotic prophase checkpoint control.

## Author summary

Genetic variation arises in sexually reproducing organisms via the combination of two processes: outbreeding and meiosis. Outbreeding ensures that individuals are unique composites of their genetically distinct parents, whereas meiosis—a specialised form of cell division—creates genetic variation within the gametes used during breeding (eggs and

(FASTQ) libraries are available via the GEO repository GSE245327.

**Funding:** LLR, MJN, DJ, WHG, GB, RMA were supported by funding from an European Research Council Consolidator Grant (311336) https://erc.europa.eu/homepage, the Biotechnology and Biological Sciences Research Council (BB/M010279/1) https://www.ukri.org/councils/bbsrc/, the Wellcome Trust (200843/Z/16/Z) and (225852/Z/22/Z) https://wellcome.org and a Career Development Award from the Human Frontier Science Program (CDA00060/2010) https://www.hfsp.org. WHG is currently supported by a Biotechnology and Biological Sciences Research Council (BBSRC) Discovery Fellowship (BB/V005081/1) https://www.ukri.org/councils/bbsrc/. The funders did not play any role in the study design, data collection and analysis, decision to publish, or manuscript preparation.

**Competing interests:** The authors have declared that no competing interests exist.

sperm in humans). During the early stages of meiosis—termed prophase—hundreds of DNA breaks are generated within parental chromosomes. Repair of these DNA breaks generates novel combinations of gene types (alleles). Despite many steps of meiosis being evolutionarily conserved from yeast to humans, it is poorly understood how DNA break formation is regulated to ensure that DNA breaks are evenly spread across all chromosomes in order to generate genetic diversity. Here, we utilise the model eukaryote, *Saccharomyces cerevisiae*, (budding yeast) to investigate the interplay between a key protein involved in DNA break recognition (Tel1, the yeast version of the mammalian gene, ATM), and the length of meiotic prophase controlled by the transcription factor Ndt80. Our observations indicate that the clustering of DNA breaks that happens when Tel1 is absent is suppressed by extending the time window of meiotic prophase, providing novel insight into the regulation of genetic variation within sexually reproducing organisms.

## Introduction

During meiosis, DNA double-strand breaks (DSBs) created by the evolutionarily conserved topoisomerase-like protein, Spo11, form in a highly regulated manner in order to initiate genetic recombination between homologous chromosomes (homologues) [1–3]. Recombination facilitates alignment of homologues along their lengths during prophase I and their subsequent accurate segregation [4,5]. Consequently, failures of either the initiation or completion of recombination can lead to chromosome segregation errors, generating inviable gametes [4,6,7].

The regulation of Spo11 activity arises at multiple levels that affect when, where, and how frequently DSBs are created across the genome [8–11]. In the sexually reproducing budding yeast *Saccharomyces cerevisiae* nine additional proteins are absolutely required for Spo11 activity, many of which have conserved functions in other species [12]. Rec102, Rec104 and Ski8 form the catalytic core with Spo11 [13,14], generating a complex with structural similarity to the ancestral heterotetrameric protein Topoisomerase VI [1,14–18]. Rec114, Mer2 and Mei4 interact with one another, bind to the structural axis of the meiotic chromosome, and are thought to regulate core-complex assembly and/or catalysis [19–22]. Finally, the evolutionarily conserved Mre11 complex (Mre11, Rad50 and Xrs2/Nbs1), has roles in both the formation and in the repair of Spo11 DSBs, the latter role performed alongside a critical repair factor component, Sae2, the orthologue of human CtIP [3,23–28]. In the absence of Sae2 (also known as Com1), DSBs accumulate with Spo11 remaining covalently bound to DSB ends via a 5′ phospho-tyrosine linkage [29–35]—consistent with Spo11's topoisomerase-like mechanism of DSB formation—enabling locus-specific and genome-wide measurements of DSB formation [36–38].

In *S. cerevisiae*, around 150–200 Spo11 DSBs are generated during the leptotene-zygotene stages of meiotic prophase, and are spread in a nonuniform manner across the four copies (two homologues, each with two sister chromatids) of the 16 chromosomes—a total of ~50 Mbp of genomic DNA [39–41]. When assayed in a population of cells, these DSBs are found to form preferentially in regions of nucleosome depletion and are termed hotspots [40,42,43]. DSB frequency within hotspots is influenced by many proactive features of the chromosome topography, including DNA replication dynamics [44–46], gene organisation [40,47], cohesin binding [48,49] and nucleosome modification [40,50,51], alongside higher-order chromosomal architectures such as centromeres, telomeres [39,40,47,52,53] and repetitive elements [40,54,55], which collectively influence the local and broad-range loading of Spo11 and other

pro-DSB factors to chromosomes [19,40,47]. In addition, greater-than-expected coincidence of Spo11-DSB formation in adjacent hotspots [56] (clustering), suggests that on a per-cell basis, subdomains of pro-DSB activity assemble upstream of DSB cleavage at different locations in different cells [10,56], but what defines and regulates their formation is unclear.

Spo11-DSB formation is also regulated reactively. As a potentially toxic DNA lesion, unrepaired DSBs are recognised by the DNA damage response (DDR) kinases Tel1 and Mec1, the *S. cerevisiae* orthologues of the human checkpoint kinases Ataxia Telangiectasia Mutated (ATM) and AT-related (ATR), respectively [57]. Tel1 activation directly inhibits further Spo11-DSB formation in a process described as DSB interference [56,58–60]. Such negative regulation appears to act relatively locally, reducing the probability of coincident DSBs arising in adjacent hotspots, a phenomenon otherwise referred to as inter-hotspot double cutting [56]. Notably, such inhibition indirectly reduces the global Spo11-DSB frequency [56,59–61], including a reduction in the formation of hyper-localised double cuts (DCs) that form within Spo11 hotspots [41,62].

Despite clear roles for Tel1 in the negative regulation of Spo11 (a conserved role carried out by ATM in mouse, plants, and flies [63–68]), the critical target(s) of the Tel1 kinase that translate such negative regulation remain obscure, with Rec114 the main lead [59,60,69]. Tel1, and its sister kinase, Mec1, also have roles in biasing DSBs to repair using the homologous chromosome and in checkpoint activation—delaying the onset of meiotic nuclear divisions as part of the DNA damage response—both via activation of the meiosis-specific Rad53/CHK2 paralogue, Mek1 [70–76]. Furthermore, down-regulation of Spo11-DSB formation is mediated via both the establishment of successful homologous chromosome interaction (termed homologue engagement [77,78]) and by the checkpoint-regulated exit from meiotic prophase via activation of the Ndt80 transcription factor [59], which is involved in the regulation of genes involved in later stages of sporulation [73,79–83]. Thus, whilst some activities of Tel1 promote meiosis-specific modes of DSB repair, contemporaneous checkpoint activation—mediated by Tel1 and others—may increase the time that cells remain in earlier stages of meiotic prophase, and thereby remain in a DSB-permissive state. However, precisely whether and how DSB interference is affected by prophase timing regulation has not been characterised.

Here, we utilise deletion of *NDT80* to explore the influence that meiotic prophase kinetics has on the process of Tel1-dependent DSB interference using both locus-specific assays and by assessing changes to the global genome-wide patterns of Spo11-DSB formation. We provide evidence that short-range DSB interference—and the manifestation of clustering—is modulated by prophase length. We further demonstrate that genome-wide patterns of DSB formation are influenced by both Tel1 and Ndt80—the latter of which we exploit to reveal chromosomal domains of preferred DSB activity.

## Results

### Deletion of *TEL1* accelerates exit from meiotic prophase in *sae2Δ* cells

We previously demonstrated that Spo11-DSB formation in *S. cerevisiae* is reactively inhibited in response to a proximal DSB, via the evolutionarily conserved PIKK kinase, Tel1 [56] (**Fig 1A**). Importantly, Tel1 has been implicated in prophase checkpoint activation in *rad50S* cells —in which DSBs accumulate without resection—causing *TEL1* mutants to exit meiotic prophase prematurely [70]. Similarly, we hypothesised that abrogation of Tel1 activity might also accelerate prophase exit (**S1A Fig**), reducing the time-window of opportunity for DSB formation in the cell populations used for our studies, which employed a deletion of the Mre11 nuclease cofactor, *SAE2*, in order to permit unresected Spo11-DSB signals to accumulate [25,26,29,31].

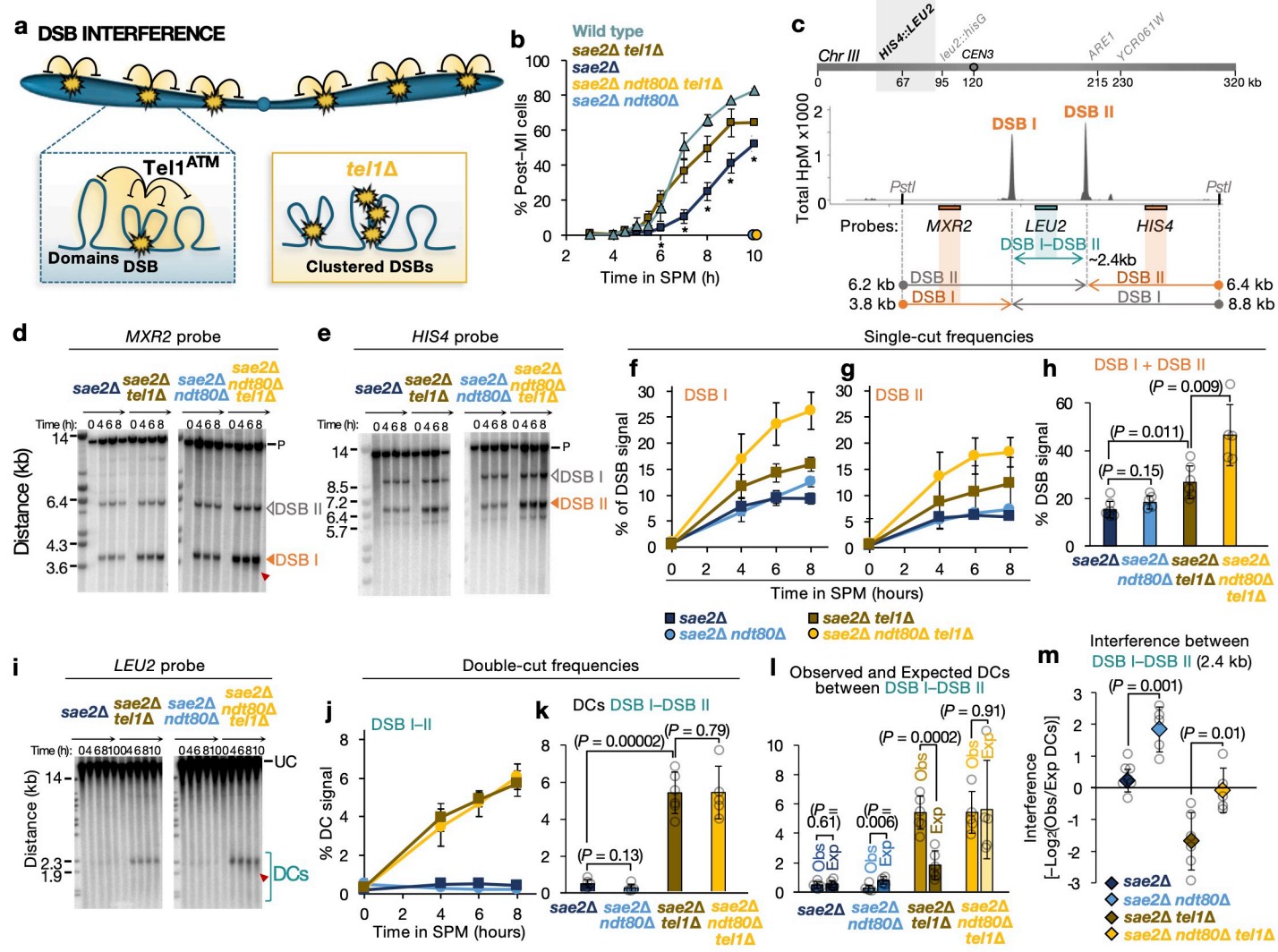

**Fig 1. Deletion of *NDT80* ablates short-range negative interference at the *HIS4::LEU2* hotspot. a**, Schematic representation of the spatial distribution of DSBs by Tel1 DSB interference in the context of the chromosome and the chromosome structure. In the absence of Tel1, the frequency of DSBs increases and DSBs are no longer subject to spatial regulation. **b**, Meiotic nuclear division (MI and MII) kinetics were assessed by counting the appearance of bi-, tri- and tetra-nucleate DAPI-stained cells. At least 100–200 cells were scored for each timepoint after inducing meiosis entry. Averages of n = 4 are represented. Asterisks indicate significant differences ($P < 0.05$) between *sae2Δ* and *sae2Δ tel1Δ* cells at the indicated timepoints. *ndt80Δ* strains were assessed only at 10 hours and 24 hours given their expected failure to exit prophase. No sporulation was observed also after 24 hours. **c**, ***Top***, Location of *HIS4::LEU2* on chromosome III. ***Bottom***, Diagram of the *HIS4::LEU2* hotspot showing Spo11-DSB positions as detected by CC-seq in hits per million (HpM; [38]), and, for Southern blotting experiments, the restriction enzyme sites, probes and size of fragments obtained from each probe. **d–e**, Representative Southern blots of genomic DNA isolated at the specified times hybridised with *MXR2* (**d**), and *HIS4* (**e**) probes. DSBs were marked with a white (non-quantified) or orange (quantified) filled triangle. Red arrow indicated DSB smear on the gels; P, *PstI* digested parental fragment. **f–g**, Quantification of DSB I (**f**), and DSB II (**g**) at the indicated timepoints. **h**, Summary of total DSBs calculated by summing DSB I and DSB II single DSBs (average of 6–8 h time points). **i**, As in **f–g** but with undigested gDNA samples at the indicated timepoints and hybridized with *LEU2* probe. Double cuts (DCs) were highlighted with a blue open bracket. UC, Uncut parental. **j**, Quantification of DC signal at the indicated time points. **k**, Summary of the observed DCs between DSB I and DSB II (average of 6–8 h time points). **l**, Quantification of observed and expected DC frequencies using averaged data from 6–8 h time points in the indicated strains. **m**, DSB interference between DSB I and DSB II calculated for each individual repeat expressed as–log₂(Observed/Expected DCs) and then averaged (see Extended methods, "Calculation of DSB interference"). In all plots, error bars indicate Standard Deviation between individual repeats (overlaid grey circles on bar graphs). For statistical analysis, a two-tailed t-test with equal variance was performed with P values indicated. n = 6 for *NDT80⁺* (4 repeats were used from Garcia et al 2015 [56] and averaged with 2 biological repeats generated in this project) and n = 5 for *ndt80Δ* backgrounds.

To test this idea, we compared meiotic prophase kinetics by monitoring the time at which cells completed the first meiotic nuclear division (MI) in synchronised meiotic cultures (**Figs 1B and S1B–S1D**). Whilst wild-type cultures started to initiate MI at ~5 hours, with MI complete in 80% of the population by 10 hours, *sae2Δ* cells were delayed by ~2–3 hours (**Fig 1B**), as previously shown [31], and similar to *rad50S* mutants [70], with very few cells proceeding through the second meiotic division by 10 hours (**S1C Fig**). By contrast—but again like in the *rad50S* background [70] —deletion of *TEL1* in the *sae2Δ* background substantially rescued these delays (**Figs 1B and S1D**).

These observations confirm that *sae2Δ* cells exit meiotic prophase earlier in the absence of Tel1, which may cause some cells to have less opportunity to initiate Spo11-DSB formation than in the presence of Tel1.

## Deletion of *NDT80* increases DSB formation in the absence of Tel1

We hypothesised that deletion of the *NDT80* transcription factor, causing meiotic cells to arrest permanently in late prophase [82] (**Fig 1B**), might equalise the length of time cells remain in prophase—and thus their DSB-forming potential—independently of the presence or absence of Tel1 activity (**S1E Fig**). For our purposes, we define "DSB-forming potential" as the pre-activation (maturation and/or priming) of sub-chromosomal regions within a cell such that they gain the potential to catalyse DSB formation. We have hypothesised that maturation involves some kind of upstream proactive, priming process [10,56]. To test this idea, we first determined the impact of extending meiotic prophase (*NDT80* deletion) on the overall frequency of DSB formation at a number of strong hotspots previously used to assess DSB interference [56]: *HIS4::LEU2* (**Fig 1C**), *ARE1* (**S2A Fig**), and *YCR061W* (**S3A Fig**).

At *HIS4::LEU2*, DSB frequency increased over time reaching a maximum at 6–8 h after meiotic induction of ~10% at site I and ~5% at site II in the *sae2Δ* control (**Fig 1D–1G**), frequencies that were not substantially altered upon *NDT80* deletion (**Fig 1F–1H**). As previously reported [56], deletion of *TEL1* in the *sae2Δ* background increased DSB frequency by around ~1.5-fold at both sites (**Fig 1D–1H**). Remarkably, however, DSB frequency was further elevated (by almost two-fold) in the *sae2Δ tel1Δ ndt80Δ* triple mutant, with total (DSB I + DSB II) levels reaching ~46% of total DNA (**Fig 1H**). Notably, DSB I signals, as measured with the *MXR2* probe, showed partial smearing down the gel suggesting a general increase in hotspot width, perhaps caused by the increase in the frequency of hyper-localised coincident cutting by Spo11 that arises within hotspots [41,62] (**Fig 1D**).

DSB frequencies measured around the *ARE1* locus were increased by deletion of *NDT80* in the *sae2Δ tel1Δ* background, reaching levels higher than those previously reported for when Ndt80 is present [56] (**S2B–S2F Fig**). At the *YCR061W* locus, the effect at individual hotspots varied (**S3B– S3F Fig**). Hotspot 'N' was increased by *TEL1* deletion, but not further by *NDT80* deletion (**S3D Fig**), whereas hotspot 'Q' was increased more by *NDT80* deletion than by *TEL1* deletion (**S3F Fig**). Notably, deletion of *TEL1* leads to the formation of a previously undetectable hotspot "O" (**S3C Fig**) flanking the *YCR061W I* probe (also detected in genome-wide CC-seq [38] maps of Spo11 DSBs; **S3A Fig**), and this hotspot was increased a further two-fold upon *NDT80* deletion (**S3E Fig**).

Collectively, such observations support the view that early exit from meiotic prophase that happens in *sae2Δ tel1Δ* cells leads to an underestimate of the total DSB potential that is possible when Tel1 is absent, and that this can be revealed by arresting cells in late meiotic prophase via *NDT80* deletion.

## Deletion of *NDT80* alters measurements of DSB interference over short range

In prior work we determined that, rather than just displaying loss of DSB interference in the absence of Tel1, over short genomic distances Spo11 DSBs were found to arise coincidently

more often than expected by chance—a phenomenon referred to as negative interference and/or clustering (**Fig 1A**, inset). We previously hypothesised that this clustering effect arises due to activation of DSB formation within a subset of meiotic chromatin loop domains [56]. Such apparent clustering can arise when an assayed population is nonhomogeneous—for example when it contains a population of active and inactive loci and/or cells—which could become especially apparent within the shortened prophase of *sae2Δ tel1Δ* cells.

Thus, to test the idea that differences in prophase length could explain our observation of negative interference and DSB clustering, we sought to re-measure DSB interference in the absence of Ndt80—which we hypothesised would increase the homogeneity of the assayed cell population. DSB interference was measured, as in our prior study (see **S1F–S1L Fig** for a description of the general method of calculation), at the *HIS4::LEU2* locus on chromosome III, in which the pair of strong DSBs are separated by just 2.4 kb (**Fig 1C**).

Interference was assessed by comparing the observed frequency of coincident DSBs ('double-cuts'; 'DCs', measured with the *LEU2* probe; **Fig 1I–1k**) to the product of the frequency (expected) of each individual DSB (DSB I and DSB II; **Fig 1L**) measured using the *MXR2* and *HIS4* probes on the left and right of the locus respectively (**Fig 1C–1H**; see Extended methods, "Calculation of DSB interference" for full description). To simplify analysis and reduce sampling error, the 6 and 8 hour time points were averaged (as in prior work [56]), and then this average value was calculated across a number of independent experimental repeats made in both the *NDT80*[+] control (n = 6) and *ndt80Δ* mutant (n = 5) backgrounds (**Fig 1H and 1K**). Measurements taken at individual time-points, whilst subject to greater technical variation, led to similar conclusions (see below).

Aggregation of additional observations made in this study with prior measurements [56] reinforced the prior conclusion: that is, in the presence of Tel1, a similar frequency of DCs were observed to those that were expected (**Fig 1L**), suggesting no interference over this short distance even though Tel1 is present (and thus the formation of DCs inhibited; **Fig 1M**). *TEL1* deletion led to a ~1.5-fold increase in the frequency of single DSBs (**Fig 1H**), but a disproportionate ~10-fold increase in the frequency of DCs (**Fig 1K**)—demonstrating not just Tel1's inhibitory role, but also how observed DCs then exceed by ~3-fold those expected by chance alone (**Fig 1L**), leading to a negative interference calculation (**Fig 1M**).

Remarkably, in the presence of Tel1—but now in the absence of Ndt80—although single DSB frequency increased a small amount (**Fig 1H**), DC frequency was unchanged (**Fig 1K**), and at a lower frequency than expected (**Fig 1L**), leading to positive interference (**Fig 1M**). Moreover, in the absence of Tel1 and Ndt80, single DSB frequencies increased further (**Fig 1H**), but without any increase in DCs relative to *tel1Δ* (**Fig 1K**), leading observed and expected frequencies of DCs to be similar (**Fig 1L**), and therefore, an absence of interference (**Fig 1M**). Analysis of individual timepoints (4, 6 and 8 hours), although expectedly more noisy (especially at 4 hours when DSB and DC signals are weaker) than when averaging the 6 and 8 hour timepoints together, reinforced these conclusions (**S1M Fig**).

To test whether similar effects were observed elsewhere, we also measured DSB and DC formation between the three main hotspots (labelled 'E', 'F', and 'I') flanking the *BUD23–ARE1* locus on chromosome III [56] (**S2A Fig**). Although other minor DSBs (and thus DCs) are also visible, their low cutting frequency and the relatively high lane background precluded their accurate measurement in this study. Deletion of *NDT80* increased single DSB frequencies in both the presence and the absence of Tel1 (**S2B–S2F Fig**) but without any major changes in DC frequencies relative to the large effect caused by *TEL1* deletion (**S2G–S2J Fig**). In agreement with the measurements made at *HIS4::LEU2* above, these effects altered DSB interference (**S2K–S2N Fig**) such that control *TEL1*[+] *ndt80Δ* cells displayed strong positive interference (**S2M and S2N Fig**; rather than weak interference), and *tel1Δ ndt80Δ* cells now displayed weak/absent interference (**S2M and S2N Fig**; rather than strong negative interference).

DSB interference measurements at a third locus (*YCR061W*) were more complicated, although displaying some similar trends (**S3 Fig**). Measuring DC formation between the main hotspot, 'N', and hotspot 'Q', 3.7 kb away (**S3G–S3I Fig**), and therein calculating DSB interference (**S3J–S3M Fig**), showed that—similar to at *HIS4::LEU2* and *ARE1*—deletion of *NDT80* when Tel1 is absent causes a substantial reduction in the negative interference previously observed [56] between hotspots N and Q (**S3L Fig**). However, potentially due to low signals and relatively high background levels (**S3G and S3H Fig**), we were unable to detect any change in interference upon deletion of *NDT80* in the presence of Tel1 (**S3L Fig**; see Extended Methods for more details). There was also no measured change in (the negative) interference detected between hotspot N and the new hotspot, O, that arises only in the *tel1Δ* background (above, **S3A and S3M Fig**)—possibly due to a combination of O being a weak, dispersed hotspot, the very short distance between hotspot N and O (~0.7 kb), and the partially overlapping probe location (see Extended Methods for more details).

In summary, whilst somewhat variable at individual loci, these observations support the view that over short distances, there is interference (mechanistically) mediated through Tel1, but that measured interference strength becomes close to zero in *TEL1*+ cells and negative in *tel1Δ* cells because of an underlying clustering of DSBs—as previously proposed [56]. Importantly, our observations here suggest that such putative effects of clustering appear to be ablated by *NDT80* deletion, leading to (the more expected) observations of positive interference in *TEL1*+ cells and little-to-no interference in *tel1Δ* cells.

## Tel1-dependent DSB interference over medium distances is unaffected by *NDT80* deletion

We next explored the impact of deleting *NDT80* on DSB interference measured over medium distances (**Figs 2 and S4**)—starting with the ~28 kb interval between the *HIS4::LEU2* and *leu2::hisG* hotspot loci inserted on the left arm of chromosome III [56] (**Fig 2A**). As measured using a probe close to the end of the chromosome (*CHA1*), average DSB frequencies at *HIS4::LEU2* were increased by *TEL1* deletion (**Fig 2B and 2C**; *P* = 0.019), but whether *NDT80* deletion alone caused an increase was unclear to due substantial measured variation (**Fig 2B and 2C**; *P* = 0.42). Deletion of both genes led to the greatest average frequency observed (~27.9%; **Fig 2B and 2C**), although, due to variation, and a relatively modest effect size (~1.2-fold increase) this was not statistically greater than the *sae2Δ tel1Δ* control (*P* = 0.19). At the *leu2::hisG* locus, the *sae2Δ ndt80Δ tel1Δ* mutant also displayed the greatest DSB frequency with, on average, a ~1.4-fold increase relative to the *sae2Δ ndt80Δ* control (**Fig 2D**; *P* = 0.011).

Whilst DCs between *HIS4::LEU2* and *leu2::hisG* (measured by the *FRM2* probe) were at or below the detection limit in the presence of Tel1, DCs were readily detected in the absence of Tel1 (*P* = 0.00002; **Fig 2E and 2F**). Deletion of *NDT80* had no detectable effect on DC formation in the presence of Tel1 (*P* = 0.47), but caused a significant ~2.2-fold increase in the absence of Tel1 (*P* = 0.00009; **Fig 2E and 2F**). Importantly, the concomitant changes in both single DSB frequencies (**Fig 2C**) and DC frequencies (**Fig 2F**) upon *NDT80* deletion, had little effect on the ratios of observed to expected DC formation in either the presence or absence of Tel1 (**Fig 2G**), and as a result no change in measurements of DSB interference between these loci (**Fig 2H**). Specifically, regardless of Ndt80 status, positive interference between these loci was retained in the presence of Tel1, but was undetectable in the absence of Tel1 (**Fig 2H**).

In agreement with these findings, measuring DSB and DC frequencies and DSB interference between the *ARE1* and *YCR061W* loci, separated by ~14 kb (**S4A–S4C Fig**), demonstrated that upon *NDT80* deletion (**S4B Fig**), DSB interference remained positive in the

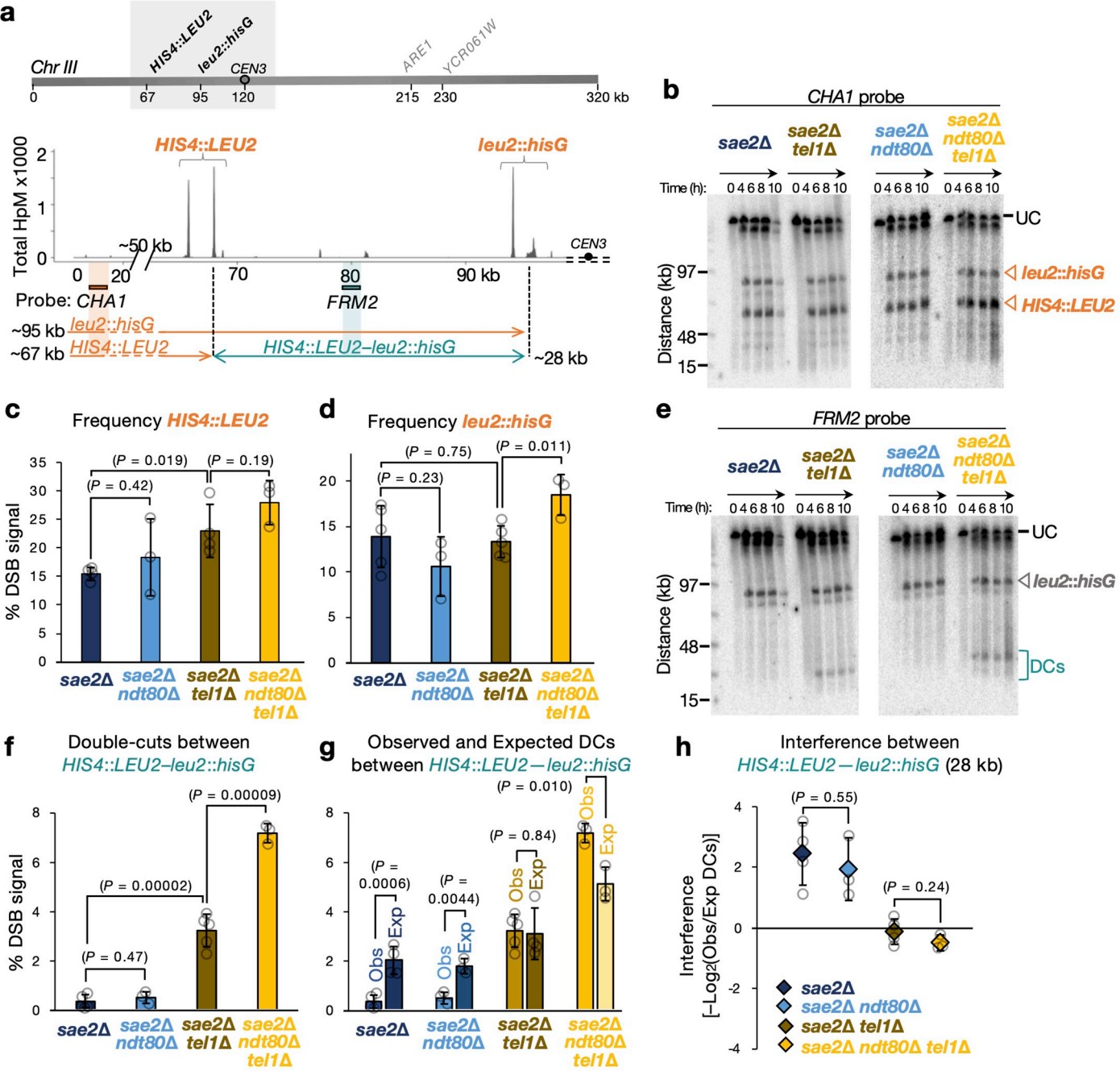

**Fig 2. Deletion of *NDT80* does not alter Tel1 DSB interference over medium distances. a**, *Top*, Location of *HIS4::LEU2–leu2::hisG* region on chromosome III. *Bottom*, Diagram of the *HIS4::LEU2–leu2::hisG* region showing positions of the DSBs as measured by CC-seq in hits per million (HpM; [38]) and, for Southern blotting experiments, the probes and size of fragments obtained from each probe. **b**, Representative Southern blots of agarose-embedded genomic DNA isolated at the specified times separated by PFGE, hybridized with *CHA1* probe. *HIS4::LEU2* and *leu2::hisG* hotspots are marked with an orange filled triangle. **c–d**, Average quantification (6 and 8 hours) of *HIS4::LEU2* (**c**) and *leu2::hisG* (**d**) hotspots. Due to the distance from the *CHA1* probe, the *leu2::hisG* DSB frequency is calculated by adding on the frequency of DCs as measured with the *FRM2* probe (as in Garcia et al 2015 [56]). **e**, As in (**b**) but hybridized with *FRM2* probe. Double cuts (DCs) between *HIS4::LEU2—leu2::hisG* are marked with a blue open bracket. UC, Uncut parental. **f**, Average quantification (6 and 8 hours) of DCs between *HIS4::LEU2—leu2::hisG*. **g**, Quantification of observed and expected DC frequencies using averaged data from 6–8 h time points in the indicated strains. **h**, DSB interference between *HIS4::LEU2* and *leu2::hisG* hotspots was calculated for each individual repeat expressed as–log₂(Observed/Expected DCs) and then averaged (see Extended methods, "Calculation of DSB interference"). In all plots, error bars indicate Standard Deviation between individual repeats (overlaid grey circles on bar graphs). For statistical analysis, a two-tailed t-test was performed with *P* values indicated. n = 5 for *NDT80*⁺ (4 repeats were used from Garcia et al 2015 [56] and averaged with 1 biological repeat generated in this project) and n = 2 for *ndt80Δ* backgrounds.

presence of Tel1 (**S4C and S4D Fig**) and remained undetectable in the absence of Tel1 regardless of Ndt80 status (**S4C and S4D Fig**).

Taken together, these observations underscore the view that Tel1-dependent DSB interference acts over both short and medium scales, but, when the length of time in meiotic prophase is limited (i.e. in *NDT80* wild-type cells), calculations of interference over short distances are altered (a skew towards lower, including negative, values) because of the underlying clustering of DSB formation that occurs in some but not all domains within the assayed population.

## Prophase arrest redistributes DSBs away from centromeric regions and regions of early Rec114 association

We recently developed covalent-complex sequencing (CC-seq), a high-resolution and genome-wide sequencing method to detect and characterise the covalent Spo11-DSB intermediates that accumulate in meiosis when *SAE2* is deleted [38] (**Fig 3A**). Based on the observations made above, we next sought to use CC-seq to explore the effects that Ndt80 and Tel1 may have on patterns (the distribution) of DSB formation at a genome-wide scale.

Taking the lead from prior work that mapped the transient Spo11-oligo intermediates liberated from DSB ends in wild-type cells [40,60], we first simplified the data into a set of ~3400 Spo11-DSB hotspots characterised by their local enrichment of reads (**S5A Fig**). The locations of these hotspots overlapped well (>85% congruence) with prior hotspot positions called from Spo11-oligo data in wild-type cells [40,60] (**S5B and S5C Fig**). Residual differences are likely caused by a combination of methodological (Spo11-oligo seq vs CC-seq) and real (*SAE2*+ vs *sae2Δ* genotypes, and presence/absence of tags on Spo11 itself) effects, and were disproportionately associated with weaker hotspots (**S5D–S5F Fig**). Notably, only a minority (32/3473; <1%) of hotspots called from the CC-seq data were also present in a *sae2Δ ndt80Δ spo11-Y135F* control sample in which the catalytic activity of Spo11 is disabled (**S5G Fig**), and these were all weak (**S5H Fig**), underscoring the utility of CC-seq for measuring bona fide Spo11-DSB formation on a genome-wide scale.

Hotspot strengths were highly positively correlated between *sae2Δ* and *sae2Δ ndt80Δ* samples (Pearson R = 0.98; **Fig 3B**), but slightly less so in the *sae2Δ tel1Δ* and *sae2Δ tel1Δ ndt80Δ* samples (Pearson R = 0.92; **Fig 3C**), suggesting again that the impact that Ndt80 has is more significant in the absence of Tel1. As expected from the highly correlated Pearson values, at broad scale, hotspot-strength distributions were visually almost indistinguishable between the four datasets when plotted along a representative chromosome (chromosome IV; **Fig 3D**). However, plotting a smoothed ratio of hotspot strength revealed spatial patterns influenced by the presence of Ndt80 that were much stronger in the absence of Tel1 (**Figs 3E and S6**).

It is important to note that these plots display fold changes in *relative* hotspot strength (Normalised hits per million mapped reads; NormHpM), which enables us to characterise the robust distributional—but not the absolute—changes in DSB strength that arise in the presence and absence of Ndt80 and/or Tel1. Current estimates carried out in *sae2Δ* cells indicate that DSB frequencies at specific hotspots increase 1.5 to 2-fold upon either Ndt80 deletion or Tel1 deletion (e.g. **Figs 1 and 2**) but how this relates to absolute changes genome-wide is still under investigation. Therefore, readers should note that where fold-change curves cross the Y-axis origin this may not indicate regions of increased or reduced DSB formation in absolute terms ±Ndt80 or ±Tel1, but rather just regions that are increased or decreased less relative to other regions.

To characterise these effects on each chromosome, ratios of hotspot strengths ±*NDT80* were represented as heatmaps binned at 50 kb scale (**Fig 3F and 3G**), and plotted centred on the centromere consistent with prior representations [84]. Effects of Ndt80 in the presence of

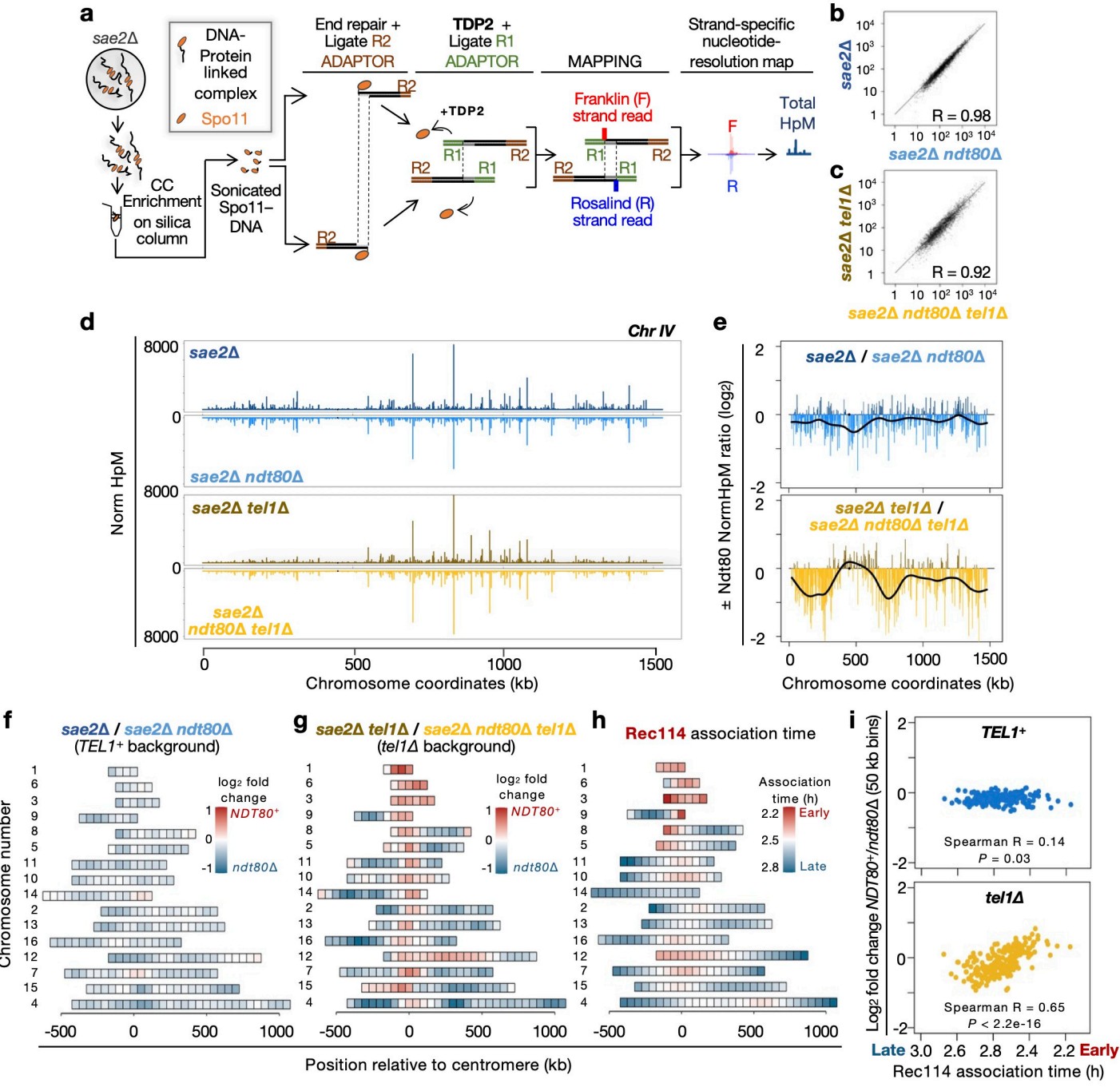

**Fig 3. Deletion of *NDT80* influences the distribution of DSBs at a genome-wide scale. a**, Schematic of the genome-wide CC-seq Spo11-DSB mapping technique (see Extended methods). **b–c**, Pearson correlation of Spo11 hotspot strengths (NormHpM) in the presence and absence of Ndt80 in *TEL1*+ (**b**) and *tel1*Δ cells (**c**). **d**, Visualization of the relative Spo11 hotspot intensities on chromosome IV in the indicated strains. **e**, Ratio of relative Spo11 hotspot intensities ±*NDT80* on chromosome IV in the presence (upper panel) and absence (lower panel) of Tel1. Values above zero indicate a higher DSB frequency in the presence of Ndt80 and below zero a higher DSB frequency in the absence of Ndt80. Fold change was smoothed to highlight the spatial trend effect of *NDT80* deletion (black line). Other chromosomes are presented in **S6 Fig**. **f–g**, Heat maps representing ±Ndt80 effect in the presence (**f**) and absence of Tel1 (**g**). Log2 ratio of relative hotspot strengths ±*NDT80* was binned into 50 kb intervals and plotted centred at the centromere and ranked by chromosome size. **h**, Pattern of Rec114 association time in hours as reported by Murakami et al (2020) [84] and presented as in **f-g**. **i**, Scatter plot of log2 fold change (*NDT80/ndt80*Δ) ±*TEL1* presented in (**f**) and (**g**) against Rec114 association time (**h**) for each 50 kb bin. The plotting order of the Rec114 data is reversed to visualise the positive relationship between early Rec114 association and regions that are enhanced in the presence of *NDT80*, an effect that is stronger in the absence of Tel1.

Tel1 were relatively modest and did not display a clear spatial pattern with respect to chromosome features such as telomeres and the centromere (**Fig 3F**). By contrast, in the absence of Tel1, the presence of Ndt80 led to a dramatic enrichment of Spo11-DSB signal in centromere-proximal regions—notably encompassing the entirety of the three shortest chromosomes (I, III, and VI), and the entire region of chromosome XII left of the rDNA array (**Fig 3G**). These observations suggest that *NDT80* deletion in the *tel1Δ* background promotes genome-wide redistribution of Spo11 activity, generating a more uniform pattern—and preventing bulk enrichment of Spo11 activity in these largely centromere-proximal regions.

To understand how this pattern of enrichment might be explained by other features of Spo11-DSB formation, we also compared our fold ratios ±*NDT80* to the time that Rec114—an essential pro-DSB factor—associates with meiotic chromosomes [84] (**Fig 3H**). Remarkably, regions of DSB formation that are enriched in the *sae2Δ tel1Δ* strain relative to the *sae2Δ ndt80Δ tel1Δ* strain (**Fig 3G**) are similar to regions that load Rec114 early (**Fig 3H**; Spearman R = 0.65, $P = 2.2\text{x}10^{-16}$; **Fig 3I**, bottom). Similarly, even though the visual effect of deleting *NDT80* in the presence of *TEL1* was very modest (**Fig 3F**) we nonetheless also detected a weak, yet statistically significant positive correlation (Spearman R = 0.14; $P = 0.03$) between regions enriched in the *sae2Δ* strain relative to the *sae2Δ ndt80Δ* strain and regions of early Rec114 association (**Fig 3I**, top). Given that Rec114 is essential for Spo11-DSB formation [3,12,20,21,85,86], our data indicate that when prophase time is limited, DSB formation is enhanced in the subset of chromosome domains in which Rec114 first associates—and that this effect then becomes particularly severe in the shorter prophase experienced by *sae2Δ tel1Δ* cells (data above). Given that *NDT80* status also alters measurements of DSB interference over short distances (above) we propose that it may be transient enrichment of Rec114 within early regions that drives the negative DSB interference (DSB clustering) that we have measured when Ndt80 is present but Tel1 is absent (above).

## Tel1 activity patterns DSB hotspot strength across the genome

We next sought to take advantage of the *NDT80* deletion-induced meiotic prophase arrest to characterise the specific genome-wide effects caused by loss of Tel1-dependent DSB interference independently of changes to prophase kinetics (**Fig 4**). Previous analysis of Spo11-oligo patterns in the presence and absence of Tel1 revealed spatially localised correlated changes in DSB hotspot strengths that decayed with distance (adjacent hotspots either went up or down in a correlated manner), with local inhibition also patterned locally by the insertion of strong DSB hotspots [60]. Globally, however, DSB hotspot strengths measured using Spo11-oligo data in the presence and absence of Tel1 are highly correlated (R = 0.97; **Fig 4A**), suggesting relatively weak global effects. By contrast, deletion of *TEL1* affected CC-seq (*sae2Δ* background) hotspot strengths more severely (R = 0.91 in *NDT80*+; **Fig 4B**), likely driven in part by the *tel1Δ*-dependent alterations in prophase length described above, and the fact that Tel1 is likely to be hyper-activated by the unresected DSBs that accumulate in the *sae2Δ* background. Indeed, even in the absence of Ndt80, when prophase kinetics are presumably equalised, CC-seq DSB hotspot strengths ±*TEL1* were less similar in the CC-seq *sae2Δ* data (R = 0.94 in *ndt80Δ*; **Fig 4C**) than in the published Spo11-oligo data ±*TEL1* (R = 0.97; **Fig 4A**; [60]).

It is important to note that in all cases, these Pearson correlation values are high, and consistent with this, like with ±*NDT80* comparisons, broad-scale hotspot-strength distributions were almost visually indistinguishable from one another between the paired ±*TEL*1 dataset comparisons when plotting along a representative chromosome (e.g. chromosome IV; **Fig 4D**). However, plotting a smoothed ratio of hotspot strengths revealed a very different picture (**Figs 4E and S7A**). Whereas effects on Spo11-oligo hotspot strength ±*TEL1* were relatively

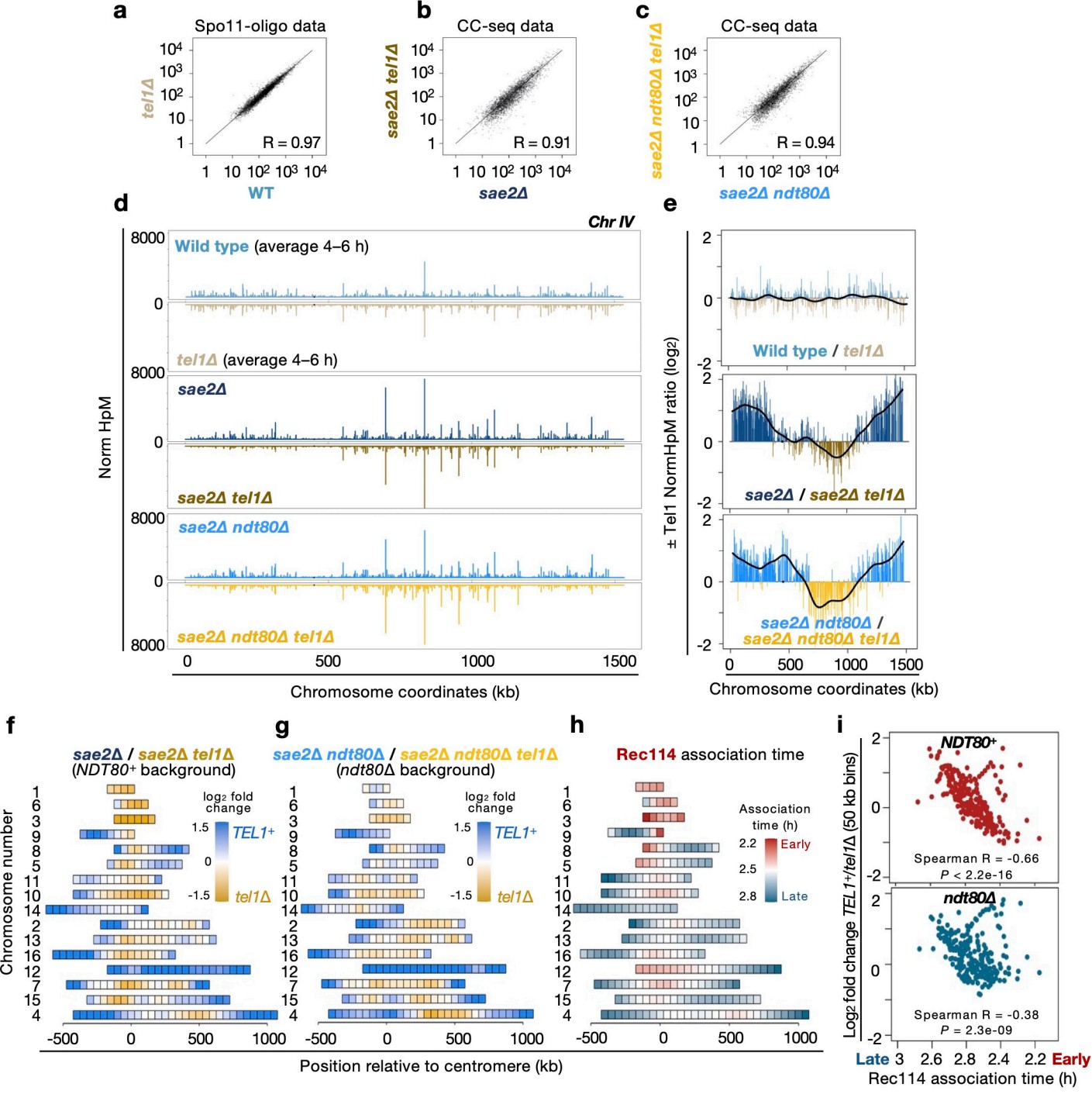

**Fig 4. Tel1-dependent genome-wide effect on DSB distribution. a–c,** Pearson correlation of Spo11 hotspot strengths (NormHpM) in the presence and absence of Tel1 in *SAE2*+ (Spo11-oligo maps; [60]) (**a**), and in CC-seq maps in *sae2Δ* (**b**) and *sae2Δ ndt80Δ* (**c**) strains. **d,** Visualization of the relative Spo11 hotspot intensities on chromosome IV in the indicated strains. **e,** Ratio of relative Spo11 hotspot intensities ±*TEL1* on chromosome IV in *SAE2*+ cells (Spo11-oligo data; upper panel) and in CC-seq maps in the presence (middle panel) and absence (lower panel) of Ndt80. Values above zero indicate a higher DSB frequency in the presence of Tel1 and below zero a higher DSB frequency in the absence of Tel1. Fold change was smoothed to highlight the spatial trend caused by *TEL1* deletion (black line). Other chromosomes are presented in **S7 Fig**. **f–g,** Heat maps of CC-seq data (*sae2Δ*) representing the ±Tel1 effect in the presence (**f**) and absence of Ndt80 (**g**). Log2 ratio of relative hotspot strengths ±*TEL1* was binned into 50 kb intervals and plotted centred on the centromere and ranked by chromosome size. **h,** Pattern of Rec114 association time in hours as reported by Murakami et al (2020) [84] and presented as in **f-g** (reproduced from **Fig 3H** to aid visual comparison). **i,** Scatter plot of log2 fold change (*TEL1/tel1Δ*) ±*NDT80* presented in (**f**) and (**g**) against Rec114 association time (**h**) for each 50 kb bin. The plotting order of the Rec114 data is reversed to match **Fig 3I**.

weak and evenly distributed (**Fig 4E**, top panel; **S7A Fig**, left column), deletion of *TEL1* in the CC-seq *sae2Δ* and *sae2Δ ndt80Δ* strains revealed strong Tel1-dependent spatially patterned chromosome-specific changes that shared similar features in both the presence and absence of Ndt80 (**Fig 4E**, middle and lower panels; **S7A Fig**, middle and right columns; **S7B Fig**). The most dramatic effects were often observed towards the ends of many chromosomes—where the distribution of DSBs was enhanced in the presence of Tel1, as was the relative proportion of DSBs forming on the entirety of chromosome XII (**Fig 4F and 4G**).

We also quantitatively compared the effect of *TEL1* deletion (**Fig 4F and 4G**) to the timing of Rec114 association (**Fig 4H**, replotted here from **Fig 3H** to enable easier visual comparison). Remarkably, regions that became enriched in the absence of *TEL1* were highly correlated with regions of early Rec114 association (**Fig 4I**), and this strong positive correlation was even stronger (Spearman R = 0.66, $P = 2.2 \times 10^{-16}$) in the presence of Ndt80 (**Fig 4I**, upper panel), than in the absence of Ndt80 (Spearman R = 0.38, $P = 2.3 \times 10^{-9}$; **Fig 4I**, lower panel). This strong positive correlation between the effect of *TEL1* deletion and early Rec114 association suggests that Tel1 is particularly important to suppress DSB formation within chromosomal regions that load Rec114 earliest.

## Discussion

We previously established in *S. cerevisiae* that Spo11 DSBs are subject to distance-dependent interference via activation of the DNA-damage-responsive kinase, Tel1—part of a negative-regulatory pathway that appears to be conserved in mice, flies and plants [63–68]. Critically, due to its involvement in the DNA damage response, Tel1 has at least two overlapping roles: DSB interference and regulation of meiotic prophase kinetics, but our understanding of how these two roles intersected was unclear and largely unexplored.

To investigate the relationship between these two roles of Tel1, we have measured the frequency of single and coincident Spo11-DSB formation arising at adjacent hotspots in the presence and absence of both Tel1 and Ndt80, the latter of which is a critical transcription factor required for exit from meiotic prophase [82]. Importantly, deletion of *NDT80* causes cells to arrest in late meiotic prophase irrespective of the strength of checkpoint activation. In order to estimate total DSB formation potential, we have utilised strains in which Mre11-dependent nucleolytic processing of Spo11-capped DSB ends is abolished via deletion of the activator, *SAE2* [25,26], permitting total DSB levels to accumulate. Importantly, due to the unresected DSBs that accumulate, Tel1's role in meiotic checkpoint control is much greater in *sae2Δ* cells than it is in wild-type cells, thereby significantly altering prophase kinetics when deleted.

When considering total DSB levels, whereas *TEL1* deletion alone tends to increase DSB formation, the effect of deleting *NDT80*—and thus of extending prophase—was generally stronger only in the absence of Tel1 (**Figs 1F–1H, S2D–S2F and S3D–S3F**), suggesting that Tel1 activity becomes even more critical in limiting DSBs in the *ndt80Δ* background. Similar estimates of global DSB formation in the presence and absence of Tel1 or Ndt80, but in the presence of Sae2, revealed increases similar to those reported here [60,77]. However, the epistatic relationship between Tel1 and Ndt80 has not been explored. Our observations suggest that Tel1 and Ndt80 likely independently limit total Spo11-DSB levels due to their separate roles in DSB interference and regulation of prophase exit, as has been discussed [59,60].

Because total DSB signals accumulate, deletion of *SAE2* also permits the analysis of instances where DSBs arise coincidentally on the same DNA molecule: "inter-hotspot double cuts". Whilst both *TEL1* and *NDT80* deletion appear, on average, to increase total DSB formation (see above), and lead to increases in the coincidence of DSB formation in hotspots that were relatively distant to one another (medium range; 20–50 kb), the same was not the case for

hotspots at close range (<15 kb). Instead, short-range suppression of double cutting largely depends only on Tel1, with modest or negligible increases upon *NDT80* deletion (**Figs 1k, S2I, S3J, S3H and S3I**).

In our prior work, we clearly demonstrated Tel1-dependent inhibition of coincident DSB formation (double cutting; [56]). Yet, intriguingly, when calculating interference, over very short inter-hotspot distances positive interference was not detected in the presence of Tel1 [56]. Moreover, when *TEL1* was deleted, coincidence of DSB formation in adjacent hotspots was higher-than-expected generating negative interference [56].

We previously proposed that these effects (the inability to detect positive interference—despite clear evidence that Tel1 impedes double cutting—and the observation of negative interference in Tel1's absence) can arise due to localised pre-activation of chromosomal domains—priming them for DSB formation in different locations in each cell [56]. For instance, although there are around ~4000 potential Spo11-DSB hotspots spread across the haploid yeast genome (totalling 16000 in the replicated diploid prophase state), only 100–200 DSBs are catalysed in any given cell (2–4 DSBs per Mbp), thus some aspect(s) of DSB formation must be rate-limiting. We propose that pre-activation of a region—upstream of DSB formation—is one such limiting step, creating—at any given chromosomal region—a heterogeneous mixture of domains that can or cannot initiate DSB formation across the cell population (**Fig 5A**, top). Critically, because in this model DSB formation is restricted to happen only within those domains that have undergone pre-activation, such heterogeneity will give rise to lower-than-expected values of DSB interference (i.e. skews towards zero or negativity; **Fig 5A**, bottom).

In wild-type cells, when Tel1 is active, the formation of such pre-activated subdomains is likely to help disperse a limited amount of DSB potential across the genome. However, in *tel1Δ*, the absence of localised negative regulation will permit efficient coincident cutting by Spo11 at all DSB hotspots located within any local pre-activated region—therein detected as negative interference [56] (**Fig 5A**, bottom).

A prediction of this subpopulation model is that any process that increases the homogeneity of the pre-activated population of domains will act to reveal the true interference strength (i.e. reduce skews towards negativity). Here we have established that skews in interference strength are abolished upon deletion of *NDT80*—suggesting that the subpopulations inferred to arise in *NDT80*[+] cells are caused by the limited time window that cells spend within meiotic prophase. Thus, in this model, the extension of meiotic prophase caused by *NDT80* deletion homogenises the population because it gives more time for chromosomal subdomains to mature and become active, reducing subpopulation effects.

Furthermore, because of the dual role of Tel1 in both DSB interference and checkpoint activation, loss of Tel1 leads to an accelerated exit from meiotic prophase (**Fig 1B**), presumably due to a relatively earlier activation of Ndt80 and subsequent down-regulation of DSB formation [59]. Such effects of Tel1 loss are likely to be more significant in the *sae2Δ* background, where DSB-dependent checkpoint activation is dependent on Tel1 [70], which is not the case under conditions where Spo11 has been removed from DSB ends and ssDNA resection has initiated [60,70]. Thus, the differential prophase timing that arises ±*TEL1* in the *sae2Δ* background very likely exacerbates the subpopulation effect. Our observations suggest that by extending the length of prophase, *NDT80* deletion can be used to limit effects caused by differential prophase kinetics, homogenising the DSB potential across the entire genome and cell population (**Fig 5B**). We contend that this is particularly important when deleting *TEL1* or other factors that influence the meiotic prophase checkpoint.

A key feature of our observations is that negative interference (and its abolition upon *NDT80* deletion) was only detected over short distances (summarised across all hotspots in

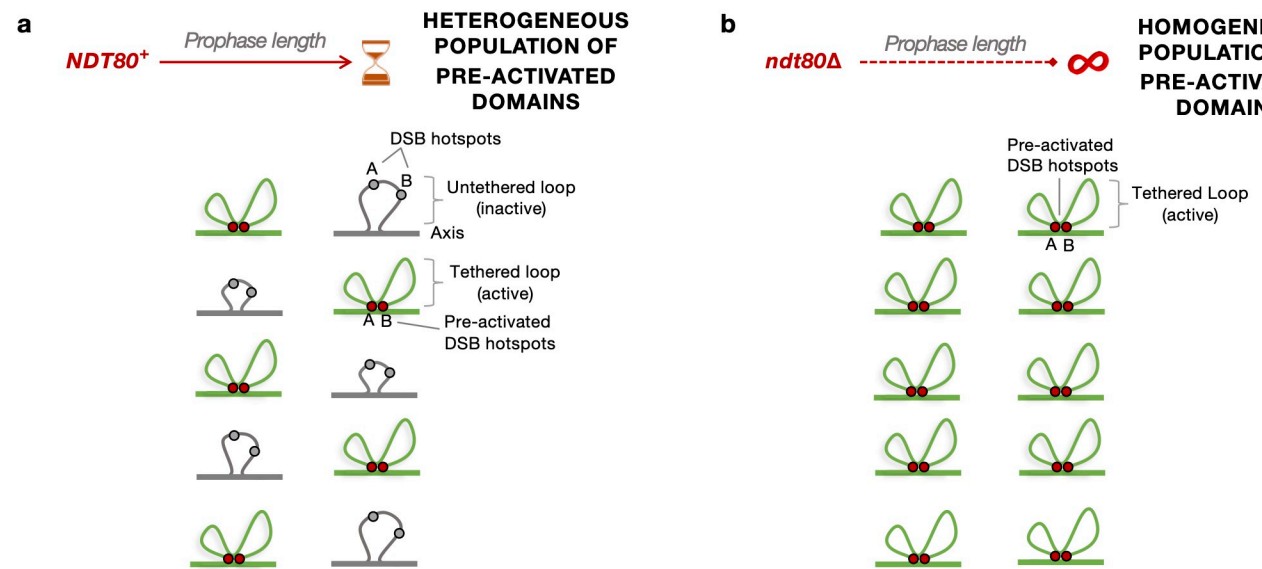

Pre-activation (or priming) refers to a process—possibly loop-axis tethering and/or some other rate-limiting step—that is a pre-requisite for DSB formation within a domain, but does not guarantee DSB formation will happen.

| **NDT80+ example:** |
|---|
| In this worked example, 50% of domains are pre-activated, enabling the potential for DSB formation to arise only within these pre-activated hotspots. |

Observed population-average frequencies at hotspots: A = 0.1 ; B = 0.05
Expected double-cut frequency: A x B = 0.005

*Probability of domain activation: 0.5

Probability of DSB formation per pre-activated domain: A* = 0.2 ; B* = 0.1
Probability of double-cut per pre-activated domain: A* x B* = 0.02

Observed population-average double-cut frequency: (A* x B*) x 0.5 = 0.01

If DSBs form independently, apparent interference: $-\log_2(0.01/0.005) = -1$

Thus, absence of interference within a domain that is pre-activated in only a subpopulation of chromosomes will result in observed negative interference proportional to: $-\log_2(1/\text{pre-activation frequency}) = -\log_2(1/0.5)$ in this example.

In wild-type cells, the presence of real interference between DSBs (e.g. via Tel1/ATM suppressing their coincident formation), will counteract this effect, pushing observed interference values to be more positive, but still lower than expected.

| **ndt80Δ example:** |
|---|
| In this worked example, the prophase arrest causes 100% of domains to be pre-activated, enabling the potential for DSB formation to arise within all hotspots. |

Observed population-average frequencies at hotspots: A = 0.1 ; B = 0.05
Expected double-cut frequency: A x B = 0.005

*Probability of domain activation: 1.0

Probability of DSB formation per pre-activated domain: A* = 0.1 ; B* = 0.05
Probability of double-cut per pre-activated domain: A* x B* = 0.005

Observed population-average double-cut frequency: (A* x B*) x 1.0 = 0.05

If DSBs form independently, apparent interference: $-\log_2(0.005/0.005) = 0$

Thus, absence of interference within a domain that is pre-activated in the entire population of chromosomes will not result in any skews in interference measurements.

In wild-type cells, the presence of real interference between DSBs (e.g. via Tel1/ATM suppressing their coincident formation), will push observed interference values to be positive.

**Fig 5. Meiotic prophase length homogenises the potential of forming active domains in which DSB formation may arise at short range. a,** Schematic representation of a heterogeneous mixture of cells with active and inactive domains with differing potential for DSB formation in *NDT80+* cells. The formation of such active/inactive subdomains will bias the measurement of DSB frequency leading to underestimates of DSB interference. In the presence of Tel1, underestimation of the coincident DSB probability within the active domains would generate what appears to be weaker interference than expected, whereas, in the absence of Tel1 (*tel1Δ*), the lack of local DSB inhibition will enable coincident cutting (DSB clustering) in the fraction of cells with the active domain, causing negative interference. In this example we represent a situation in which 50% of the assayed population have the domain pre-activated at the tested region. **b,** We propose that deletion of *NDT80* extends the length of the meiotic prophase homogenising the potential for domains to be pre-activated and allowing a more accurate estimate of DSB frequency per active domain. In the presence of Tel1, localised inhibition will cause DSBs to arise more evenly across the genome—leading to detection of positive interference, whereas in *tel1Δ* cells, the lack of inhibition will lead to detection of no interference. In this example we represent a situation in which 100% of the assayed population have the domain pre-activated at the tested region. Although Spo11-DSB formation arises in the context of a maturing loop-axis chromosome structure organised by cohesin, and contains chromatin loops that are within the size range (in *S. cerevisiae*) over which we infer pre-activation to occur (<15 kb), such pre-activated domains may simply coincide with, and co-occur alongside loop formation, but not necessarily depend upon their existence, instead being driven by the assembly of pro-DSB factors such as Rec114, Mei4 and Mer2 (RMM; see discussion for further details).

S4E Fig)—behaviour that is consistent with zones of activation being of relatively limited size (<15 kb). Although Spo11-DSB formation arises in the context of a maturing loop-axis chromosome structure organised by cohesins [87], and contains chromatin loops that are within this size range in *S. cerevisiae* [88,89], such active domains may simply coincide with and co-occur alongside loop formation, but not necessarily depend upon their existence (see pro-DSB section below).

We have also explored the changes in genome-wide patterns of Spo11-DSB formation that arise in the presence and absence of Ndt80, and how these differences are affected by *TEL1* deletion (Fig 3). Importantly—and consistent with our hypothesis that accelerated exit from prophase in *sae2Δ tel1Δ* accentuates the impact of subpopulation domains in which Spo11 is active—absence of *NDT80* led to a much stronger change in the genome-wide pattern of DSB formation in *sae2Δ tel1Δ* cells than in *sae2Δ* cells (Fig 3F and 3G). Such a difference is expected due to the more limited temporal window of meiotic prophase that otherwise arises in the absence of Tel1.

Critically, when Tel1 is absent, regions where Spo11 activity is greatest in the presence of Ndt80 correlate with regions that load Rec114 and Mer2 early in meiosis [84] (Fig 3G and 3H), arguing that when the temporal window of meiotic prophase is limited, DSBs tend to arise more often in those regions that load pro-DSB components more efficiently. By contrast, when the duration of meiotic prophase is extended (by *NDT80* deletion), DSBs now arise more evenly across the genome—with a disproportionate relative enhancement in regions that load pro-DSB factors late. It has been suggested that DSB distributions and frequencies arising in *sae2Δ* or *rad50S* cells represent an underestimate of the wild-type patterns [44]. However, given that deletion of *NDT80* had only a relatively minor impact in the presence of Tel1 (Fig 3F) suggests that any underestimate is likely not due to premature exit from meiotic prophase.

In general terms, we propose that it is the disproportionate loading of pro-DSB factors in some genomic regions that drives the negative DSB interference (DSB clustering) detected over short distances upon *TEL1* deletion [56]. Precisely how Rec114, Mer2 and Mei4 regulate DSB formation remains to be elucidated, however, their potential to form limited amounts [20,21,90] of intermolecular condensates [22], which may generate a surface for DSB formation [22,41], makes them prime candidates for generating pre-activated domains of local DSB potential.

A second finding that emerges from our genome-wide studies, is that despite the influence that temporal changes in meiotic prophase timing (i.e. *NDT80* deletion) has on DSB distribution (Fig 3F and 3G), deletion of *TEL1* elicits a much stronger effect on the DSB distribution, which is detectable both in the presence and absence of Ndt80 (Fig 4F and 4G). We hypothesise that at least one component underpinning these strong Tel1-dependent changes is the genome-wide effect of DSB interference (LLR & MJN, manuscript in preparation), which is robust to changes in the length of meiotic prophase. Why DSBs on chromosome XII are particularly dependent on Tel1, however, is intriguing and is not predicted by a simple interference model. Given the presence of the ~1 Mbp ribosomal DNA (rDNA) array on chromosome XII, and the known genetic interactions between Tel1, Sae2, and Pch2 [91–95]—the latter of which is a remodeller of the pro-DSB axial factor Hop1 [96–98], is involved in checkpoint activation itself [91,92,99], and is both localised to, and important to suppress DSBs adjacent to, the rDNA region [54]—we speculate that Pch2 may be involved.

A limitation of our analytical methods is the reliance on *SAE2* deletion to permit Spo11-DSB and Spo11 double-cut signals to accumulate without repair. On the one hand, *sae2Δ* enables us to study mechanisms of DSB interference in the absence of other regulatory pathways that are dependent upon and triggered after Spo11 removal (i.e. homologue engagement [77,78]), and which may otherwise obscure Tel1's influence. Yet, we cannot exclude that the accumulation of unrepaired Spo11 DSBs itself influences how the system behaves. For

example, if like in mitotically dividing cells [70,100] Tel1 is hyperactivated upon *SAE2* deletion, then the magnitude of the effects on DSB interference and DSB distributions that arise upon deleting *TEL1* are expected to be stronger than they would otherwise be in wild-type cells where *SAE2* is present. Indeed, as has been presented [60] (and within our study, above), genome-wide effects of deleting Tel1 in wild-type cells are relatively modest as assessed from patterns of Spo11-oligos. One interpretation of this difference compared to in the *sae2Δ* background, is that Tel1 may have a rather limited temporal and/or spatial capacity to inhibit adjacent DSB formation in wild-type cells. By contrast, the persistence of DSBs—and thus presumably also the persistence of Tel1 activation that arises—in *sae2Δ* cells causes much stronger spatial patterning. As such, the observations presented here should be interpreted with such differences in mind.

Our study was also limited to measuring DSB interference only between the strong hotspots on chromosome III. Whilst attempts to characterise DSB interference at other genomic loci were made (as performed in prior work [56]), the resulting data displayed too much technical variation to enable useful analysis. It is therefore important to recognise that the DSB interference measurements—and the changes we observe upon *TEL1* and *NDT80* deletion—are limited to regions of the genome that, based on Rec114 association timing [84], are likely to generate DSBs earlier than the genome average. However, whether it is the Rec114 timing difference itself—or some other feature of DSB regulation—that underpins the inferred subpopulation timing effects will require further investigation. For example, it is reasonable to expect that genomic regions that load Rec114 later are also (or even more) subject to subdomain effects due the fact that relatively few cells in the population may have an opportunity to prime such genomic regions for DSB formation.

Looking more broadly, the regulatory feedback mechanisms discussed here are likely to ensure that cells stay in a DSB-permissive state only for as long as needed—limiting the level of DSB formation, and therefore recombination, required to facilitate accurate chromosome pairing and, by extension, efficient chromosome segregation without risk of aneuploidy. Because of Ndt80's role as a transcription factor we favour that the effect Ndt80 elicits is global, influencing the length of time any individual cell remains in meiotic prophase. However, it is also possible that targets of Ndt80 act locally to suppress and inhibit Spo11 activity, directly creating heterogeneity in which chromosomal regions are active within individual cells. It is also possible that the additional DSBs that form in the absence of Tel1 when *NDT80* is deleted do not display negative interference because clustering is suppressed by a factor that acts redundantly with Tel1, but only at later stages of meiotic prophase. Based on studies of recombination control in wild-type cells, one such candidate could be the delayed activation of the ATR orthologue, Mec1 [95]. Regardless of mechanism, our observations highlight how restrictions on global Spo11 activity can generate subdomains of concerted activity—influencing both localised and population-average patterns of genetic recombination.

## Material and methods

### Yeast strains

All the *Saccharomyces cerevisiae* yeast strains used in this study are in the SK1 background as described in **S1 Table**, and derived using standard techniques. Strains contained the *his4X*::*LEU2* and *leu2*::*hisG* exogenous sequences inserted on chromosome III [38,41], and carried the *ndt80Δ*::*LEU2*, *tel1Δ*::*HphMX4* and/or *sae2Δ*::*kanMX* gene disruption alleles [56,74,82,101]. The *spo11-Y135F*::*KanMX* allele contains an inactivating mutation of the catalytic tyrosine residue [16].

## Culture methods

For meiosis induction, a single colony was inoculated in 4mL of YPD medium (1% yeast extract, 2% peptone, 2% glucose supplemented with 0.5 mM adenine and 0.4 mM uracil) and incubated at 30˚C, 250 rpm for a day to reach saturation, then diluted to OD600 of 0.2 in a volume of 200 mL of either YPA (1% yeast extract, 2% peptone, 1% potassium acetate) or SPS (0.5% yeast extract, 1% peptone, 0.67% Yeast Nitrogen Base without amino acids, 1% potassium acetate, 0.05M Potassium Hydrogen phthalate, 0.001% Antifoam 204) pre-sporulation medium. Cultures were incubated at 30˚C, 250 rpm for 14–16 hours, then washed and resuspended in 200 mL pre-warmed SPM sporulation medium (2% potassium acetate supplemented with diluted amino acids) and incubated at 30˚C, 250rpm for the duration of the time course. Samples were taken at the relevant timepoints and processed differently. For DNA extraction, 20 mL of culture was taken at t = 0, 4, 6 and 8 hours after inducing meiosis. Samples were centrifuged at 3000 x $g$ for 4 minutes, supernatant was discarded and pellet resuspended in 2 mL 50 mM EDTA, centrifuged again for 1 minute at 3000 x $g$, supernatant discarded and pellet stored at -20˚C until use. For Spo11 CC-seq, 50 mL of culture was taken at t = 6 hours. Samples were centrifuged at 3000 x $g$ for 5 minutes, supernatant discarded and pellet frozen at -20˚C until use. For FACS, 200 μL of culture was taken at t = 0, 2, 4, 6 and 8 hours after inducing meiosis, samples were centrifuged at 16,000 x $g$ for 1 minute, supernatant discarded, fixed in 1mL of 70% EtOH and stored at 4˚C until use. For DAPI staining, 195 μL of culture was taken at t = 3, 4, 5, 5.5, 6, 7, 8, 9 and 10 hours after inducing meiosis. Cells were fixed in 450 μL of 100% EtOH and stored at -20˚C until use.

## FACS

Samples were centrifuged at room temperature, 16,000 x $g$ for 1 minute. Supernatant was aspirated, pellet resuspended in 500 μL 10 mM Tris HCl pH 8.0 / 15 mM NaCl / 10 mM EDTA pH 8.0 / 1 mg/mL RNase A and incubated at 37˚C for 2 hours at 800 rpm on a Eppendorf Thermomixer. Samples were then centrifuged at 16,000 x $g$ for 1 minute, supernatant aspirated, pellets resuspended in 100 μL of 1 mg/mL Proteinase K + 50 mM Tris HCl pH 8.0 and incubated at 50˚C for 30 minutes at 800 rpm on a Eppendorf Thermomixer. Samples were centrifuged and supernatant aspirated. Pellets were washed in 1 mL 1M Tris-HCl pH 8.0 and then resuspended in 1 mL 50 mM Tris-HCl pH 8.0 + 1 uM Sytox green. Samples were stored overnight at 4˚C and then sonicated at 20% amplitude for 12–14 seconds before being sorted by flow cytometry (Accuri Flow Cytometers).

## Cell fixation and DAPI staining

Ethanol-fixed cells (4 μL) were dried at RT on a glass slide, stained with 2 μL of Fluoroshield DAPI Sigma-Aldrich (F6057-20ML) and 100–200 mono-, bi-, tri-, tetra-nucleate cells were scored by microscopy (Zeiss AXIO) using fluorescence (CoolLED pE-300 lite). Meiotic progression was determined based on the frequency of cells that entered MI (binucleated) or MII (tri-, tetra-nucleate) at different timepoints after inducing meiosis.

## Proteolytic gDNA extraction

Meiotic cell culture pellets were defrosted at room temperature, resuspended in 500 μL of spheroplasting mix: 492.5 μL of spheroplasting buffer (1 M sorbitol / 100 mM NaHPO4 pH 7.2 / 100 mM EDTA), 2.5 μL of zymolyase 100T (50 mg/mL) and 5 μL of β-mercaptoethanol, and incubated at 37˚C for 1 hour. Cells were lysed by adding 100 μL of 3% SDS / 0.1 M EDTA plus 5 μL of Proteinase K (50 mg/mL), and incubated overnight at 60˚C. After cooling to room

temperature, proteins were removed with 500 μL of phenol/chloroform: two rounds of vigorous shaking separated by a 5-minute rest and followed by a 5 minute centrifugation at 14,000 rpm. DNA and RNA were extracted from 450 μL of the aqueous phase and precipitated with 45 μL of 3 M NaAc pH 5.2 and 500 μL of 100% EtOH, centrifuged at 14,000 rpm for 1 minute, aspirated and washed with 1 mL 70% EtOH, pulsed down, air dried for 10 minutes and resuspended in 450 μL of 1x TE (10 mM Tris / 1 mM EDTA pH 7.5) overnight at 4˚C. RNA was digested with 50 μL of 1 mg/mL RNase A (10 mg/ml stock) for 1 hour at 37˚C. DNA was precipitated by addition of 50 μL of NaAc pH 5.2 and 1 mL of 100% EtOH, mixed by inversion and centrifuged for 1 min at 14,000 rpm. DNA was washed with 1 mL 70% EtOH, pulsed down, air dried for 10 minutes, dissolved in 200 μL of 1x TE (10 mM Tris / 1 mM EDTA pH 7.5 prep room solution) overnight at 4˚C. To measure the frequency of DSBs (single-cuts) gDNA was digested with a restriction enzyme (as described in **S2 Table**, digestion column). For 20 μL of gDNA, 6 μL of H2O, 3 μL of enzyme buffer and 1 μL of enzyme was added and incubated overnight at 37˚C. To quantify double DSB events (double-cuts), gDNA was left undigested (**S2 Table**).

## DSB analysis by Southern blot

0.7% or 0.8% agarose gels were prepared for digested and undigested gDNA samples, respectively. The gel was mixed with 125 μL EtBr (0.1 mg/mL) and allowed to set for 1 hour at room temperature. 20 μL of digested sample + 1x loading dye or 10 μL of gDNA + 10 μL water and 1x loading dye was loaded on wells. For the ladder, 10 μL of Lambda BstE II-digest was used. DNA was separated at 45–50 V for 15–19 hours. Gels were imaged using the Syngene InGenius bioimaging system. DNA was nicked by exposure to 1800 J/m2 UV in a Stratalinker. Afterwards, the gel was soaked in denaturing solution (0.5 M NaOH, 1.5 M NaCl), on a shaker for ~30 minutes.

## DSB analysis by PFGE

DNA was embedded in agarose plugs as described below. Agarose plug preparation: Cell pellets were defrosted at room temperature and washed twice with 50 mM cold EDTA (resuspended, spun 1 minute at 4˚C 3000 x g and aspirated). Cells were then resuspended with 135 μL of solution 1 (50 mM EDTA + SCE [Filtered 1 M sorbitol, 0.1 M sodium citrate, 0.06 M EDTA pH 7] + 2% BME + 1 mg/mL zymolyase 100T) and 165 μL of pre-warmed 1% LMP agarose (1% agarose in 0.125M EDTA) at 55˚C. The mix was cooled down at 4˚C for 30 minutes. The solidified plugs were added onto 1 mL of solution 2 (0.45 M EDTA + 20 mM Tris-HCl pH 8 + 1% BME + RNase 10 ug/mL + water) and incubated for 2 hours at 37˚C. Samples were inverted every 30 minutes. Solution 2 was aspirated, and plugs were covered with 1 mL of solution 3 (0.25 M EDTA + 20 mM Tris-HCl pH 8 + 1% sodium sarcosine + 1 mg/mL proteinase K + water) and incubated overnight at 55˚C. Solution 3 was aspirated and samples washed three times with 1 mL of 50 mM EDTA on a rotary wheel. EDTA was aspirated and plugs were covered by 1 mL of storage buffer (50 mM EDTA, 50% glycerol) and stored at -20˚C until use. PFGE gel: 1.3% agarose gel was prepared using 150 mL of 0.5X TBE (diluted from 5X TBE: 450 mM tris base + 450 mM boric acid + 10 mM EDTA pH8 + water) and cooled down to 55˚C before use. The plugs were cut in half and washed in 2.5 mL of 0.5X TBE on the rotary wheel for 15 minutes. The plugs were loaded in order onto the gel comb and fixed with 1% agarose. A slice of mid-range PFG marker (#3425, NEB) was fixed onto the first and last gel combs. Once set (10 min at room temperature), the 1.3% agarose was poured covering the gel combs and allowed to solidify for 30 minutes. The gel comb was removed, and wells filled with 1% agarose and allowed to set (10 min at room temperature), then immersed into pre-cooled

0.5X TBE buffer for 15 minutes. For DNA fragments of ~150 kb, the gel was run 30–30s for 3 hours + 3–6s for 37 hours at 6 V/cm and 120˚C angle. After electrophoresis, the gel was soaked in 150 mL distilled water + 7.5 µL of EtBr (0.1 mg/mL), shaken for 20 minutes and then imaged using the Syngene InGenius bioimaging system. DNA was nicked by exposure to 1800 J/m2 UV in a Stratalinker, then soaked in a denaturing solution (0.5 M NaOH, 1.5 M NaCl) whilst shaking for ~30 minutes.

## Southern blotting transfer and hybridisation

The denatured gels were transferred to a Biorad Zeta-probe membrane under vacuum (50–55 mBar for ~2 hours) in 0.5 M NaOH, 1.5 M NaCl. The membrane was washed twice with 2x SSC (diluted from 20x: 3M Sodium chloride, 0.3 M trisodium citrate pH 7.0) and then thrice with distilled water. The membrane was cross-linked by exposure to 1200 J/m2 UV in a Strata-linker, dried at room temperature for 1 hour and stored at 4˚C until probed. Southern Blot membranes were incubated with 35 mL of a pre-warmed hybridisation solution (0.5 M NaHPO$_4$ pH 7.5, 5% SDS, 1 mM EDTA, 1% BSA) at 65˚C for ~1–2 hours. To quantify the single and double DSBs, the membranes were hybridised with an appropriate DNA probe—as indicated in figure legends and **S2 Table**—radiolabelled with P$_{32}$ prepared via random priming using the High Prime kit (BioRad). Pre-hybridisation solution was discarded, and the membrane incubated with 20 mL of hybridisation solution containing the radioactive probe overnight at 65˚C. After incubation, the membrane was washed (10% SDS / 1M NaHPO$_4$ / 0.5 M EDTA), air dried, and exposed to a phosphor screen overnight. After exposure (usually 8–48 hours), the phosphoscreen was scanned with a Fuji FLA 5000 reader and analysed using Ima-geGauge software (Fuji). DSB and DC quantification methods, and limitations of the technique are described in **Extended Methods**.

## Covalent complex sequencing (CC-seq) mapping

Protein-DNA Covalent-Complex Mapping (CC-seq) in yeast followed a method previously described [38]. Briefly, meiotic cell samples are chilled and frozen at -20˚C for at least 8 hours, then thawed and spheroplasted (in 1 M sorbitol, 50 mM NaHPO4, 10 mM EDTA, 30 min at 37˚C), fixed in 70% ice-cold ethanol, collected by centrifugation, dried briefly, then lysed in STE (2% SDS, 0.5 M Tris, 10 mM EDTA). Genomic DNA was extracted via Phenol/Chloro-form/IAA extraction (25:24:1 ratio) at room temperature, with aqueous material carefully collected, precipitated with ethanol, washed, dried, then resuspended in 1xTE buffer (10 mM Tris/1 mM EDTA). Total genomic DNA was sonicated to <500 bp average length using a Covaris M220 before equilibrating to a final concentration of 0.3 M NaCl, 0.1% TritonX100, 0.05% Sarkosyl. Covalent complexes were enriched on silica columns (Qiagen) via centrifugation, washed with TEN solution (10 mM Tris / 1 mM EDTA / 0.3 M NaCl), before eluted with TES buffer (10 mM Tris / 1 mM EDTA / 1% SDS). Samples were treated with Proteinase K at 50˚C, and purified by ethanol precipitation. DNA ends were filled and repaired using NEB Ultra II end-repair module (NEB #E7645), with adapters ligated sequentially to the sonicated, then blocked, ends with recombinant TDP2 treatment in between these steps to remove the 5-phos-photyrosyl-linked Spo11 peptide. Ampure bead cleanups were used to facilitate sequential reactions. PCR-amplified libraries were quantified on a Bioanalyser and appropriately diluted and multiplexed for deep sequencing (Illumina MiSeq 2x75 bp).

FASTQ reads were aligned to the reference genome (SacCer3H4L2; which includes the *HIS4*::*LEU2* and *leu2*::*hisG* loci inserted into the Cer3 *S. cerevisiae* genome build [38,41]) via Bowtie2, using TermMapper as previously described [38,41] (**https://github.com/Neale-Lab/terminalMapper**), with all subsequent analyses performed in R version 4.1.2 using RStudio

(Version 2021.09.0 Build 351). Reproducibility between libraries for independent biological replicates was evaluated and validated prior to averaging. For detailed information see **Script summary description**. Details of individual libraries are presented in **S3 Table**.

### Calibration of CC-seq libraries

For each library the proportion of non-specific reads (background reads) were estimated by measuring the hit rate per million reads per base pair (HpM) in 47 of the longest gene ORFs (> 5.5 kb long) in the *S. cerevisiae* genome. For detailed information about the mechanics of the script, see Calculating background reads.R in **https://github.com/Neale-Lab/Ndt80_LLR**.

### Bioinformatic analysis of Spo11-DSB

All bioinformatics analyses were performed in R (R version 4.1.2) using RStudio (Version 2021.09.0 Build 351). Scripts are available on **https://github.com/Neale-Lab/Ndt80_LLR**. For further information see **Script summary description.** To compare CC-seq maps with Rec114 association timing data [84], these data (GSE52970_tRec114_A+) were downloaded, filtered to remove low confidence points as indicated by the authors, smoothed using the loess function (80 divided by the number of total points in order to normalise for chromosome size), then segmented into 50 kb non-overlapping bins.

### Extended methods

#### DSB and DC quantifications

DSBs and DCs were quantified with Image gauge software (Fuji). The DSB profile was defined by drawing lanes from the base of the parental band down to the end of the last quantifiable DSB. Background signal was manually removed with a linear subtraction. The signal above the threshold was quantified as a specific signal. DSBs and DCs were quantified as a fraction of the total lane signal observed on gel (which included uncut parental plus all the visible bands). Bands that were observed at time = 0 hours were considered nonspecific and thus not quantified. As previously described [56], at the *HIS4::LEU2* locus, the fraction of DCs detected with the *LEU2* central probe was multiplied by 3 in order to correct for the fact that only a third of the detected parental DNA signal is derived from the *HIS4::LEU2* locus because these strains contain three copies of the *LEU2* gene (*his4X::LEU2*, *leu2::hisG* and *nuc1::LEU2*). For *ARE1* and *YCR061W* hotspots, the main hotspot (F and N, respectively) was measured with an adjacent probe on the side where there were fewer DSBs prior to the main hotspot, and then corrected by adding the double cuts event present at that region. In the case of *ARE1*, F was measured from the right using *PWP2* probe and the value corrected by adding FI double cuts measured with *ARE1* probe. In the case of *YCR061W*, the main hotspot, N, was measured from the right using *YCR061W II* probe and corrected by adding NO double cuts measured with *YCR061W I* probe. Similarly, at *HIS4::LEU2–leu2::hisG* loci, quantification of the *leu2::hisG* hotspot was measured using *CHA1* probe and corrected by adding *HIS4::LEU2–leu2::hisG* double-cuts measured with *FRM2* probe. Quantification of the single or double DSB events were displayed as an average of 6, 8 (and occasionally 10) hours after meiosis induction from each repeat. The number of biological repeats is indicated in figure legends. For **Figs 1 and 2**, measurements from the $NDT80^+$ background were an average of the data published by Garcia et al (2015) [56] and one and two extra biological replicates developed in this analysis (as specified in figure legends). For **S2–S4 Figs**, measurements from the $NDT80^+$ background came only from the data published by Garcia et al (2015) [56].

## Calculations of DSB interference

To study DSB interference between two hotspots, the observed frequency of double DSB events that arise at the same molecule (observed DCs) was compared with the expected frequency on the assumption of independence (expected DCs) as described in **S1F–S1J Fig**. Such expected DC frequency was estimated from multiplication of the frequencies of single DSB events between which DSB interference is studied. To study the strength of interference, the coefficient of coincidence (CoC) was estimated by dividing the observed frequency of DCs by the frequency of expected DCs. Whilst prior studies have subtracted this value from 1 and represented the result in linear mathematical space [56], we instead performed a–$\log_2$ transformation that has the benefit of retaining positive and negative values of interference, but also corrects the unbalanced skews that arise when a ratio is presented in linear space (**S1K Fig**). As before, positive values suggest positive interference, values close to zero suggest independence (no interference) and negative values suggest concerted DSB activity (negative interference; **S1L Fig**). DSB interference was calculated on a time course basis—using the time 6 and 8 h averaged frequencies to reduce technical variation—and then averaged across all time courses.

As an example, at *HIS4::LEU2*, the frequency of DCs between DSB I and DSB II was measured with a central probe *LEU2* (**Fig 1C and 1I**). For each repeat, the averaged observed DCs from time 6 and 8 hours, was then compared with the expected frequency of coincident cuts (also averaged time 6 and 8 hours) obtained by multiplying the averaged frequency of DSB I (measured with *MXR2* probe) and DSB II (measured with *HIS4* probe) (**Fig 1C–1E, 1L**). Strength of interference was calculated as:–$\log_2$ [(Averaged observed DC DSB I–DSB II) / (Averaged expected DC DSB I–DSB II)] in several biological repeats (n = 6 in the case of *NDT80*+ background and n = 5 in the case of *ndt80Δ* background) (**Fig 1M**). Using this method, one interference measurement was produced for every repeat and then averaged. Standard deviation was calculated and a two-tailed t-test performed to assess significant differences between the strains as indicated in the figure legends.

## Limitations of the Southern blot and pulsed-field gel electrophoresis techniques

Due to the requirement of multiple gels and the limited resolution of Southern blots and PFGE methods, DSB and DC quantifications are best estimates given the following technical limitations. First, the analysis of many gels analysed in this study indicated that the magnitude of the values varies between both biological and technical repeats depending on gel/blot quality. Second, the strength of the band signal highly influences the quantifications. For instance, weak hotspots are more difficult to characterise than strong ones. Moreover, quantification of DC molecules is challenging in the *TEL1*+ background because the level of coincident DSBs is low and generally at or below the detection limit, therefore most of the signal measured is background signal which sometimes can be higher than the calculated expected random frequency (if any of the hotspots is weak) and thus leading to an underestimate of interference strength (e.g. hotspot N and Q; **S3G and S3H Fig**). Finally, because the strength of DSB interference is calculated using the division of the observed frequency of double-cut molecules by the frequency expected from independence, the result may be inaccurate when the observed and expected values are close to zero because it produces a disproportionate relative difference that may be artefactual. For example, for this reason the strength of interference was excluded between NO at the *YCR061W* hotspot in the *sae2Δ* and *sae2Δ ndt80Δ* (**S3M Fig**).

Another limitation of these techniques is that they only permit an estimate of the number of broken chromatids and not how many times a chromatid has been broken, therefore, quantification of the total frequency of DSBs may be underestimated if the frequency of double

events is high (as is the case of *tel1Δ* mutants). Furthermore, the direction and distance from the probe to the hotspot also influences the accuracy of hotspot detection. For example, due to hotspots having a width of 100–300 bp, when the inter-hotspot distance is very short (e.g. hotspots NO, 0.7 kb apart; **S3A Fig**), DC sizes are more variable as a proportion of their length, and the DC probe may also overlap with the DSB positions—both of which may affect their detection on gels.

On the other hand, when the distance between the probe and the measured hotspot is large, the presence of hotspots close to the probe will cause an underestimate of the real frequency of DSBs that are further away. For instance, quantification of the main *ARE1* hotspot "F" slightly differs when measured from the right side of the DSB using *PWP2* probe or from the left with the *TAF2* probe (**S2 Fig**). In this example, measurement of F with *TAF2* reported a lower amount of F than *PWP2* probe probably due to the presence of the strong hotspot E prior to F, thus closer to the *TAF2* probe.

The location of the probe is also another factor to consider, especially when a DSB only arises concertedly with another DSB. This seems to be the case of the band smear at *HIS4*::*LEU2* locus detected with *MXR2* and *LEU2* probe but not *HIS4* probe in *sae2Δ ndt80Δ tel1Δ* mutants (**Fig 1D and 1I**, red arrow). In fact, the smear detected with the *LEU2* probe indicates the presence of shorter DCs, which would be consistent with either DSB I or DSB II, or both, cutting in different positions within the DSB I–DSB II region. The fact that we can only observe spreading from one side (using *MXR2* probe) indicates that such alternative cutting only happens when DSB I and DSB II cut coincidently, and thus the presence of DSB I will obscure DSB II if measured with *HIS4* probe. Despite these caveats, Southern-blotting techniques are a highly valuable tool to estimate DSB interference at specific loci.

## Hotspot identification

Due to the potential variations in the DSB formation displayed in some of the mutants used in this research, a new template of hotspot coordinates was developed in a similar manner to that described in Pan et al (2011) [40] (**S5A Fig**). Hotspots were identified in our baseline *sae2Δ ndt80Δ* strains, as well as in other pertinent mutants using a 201 bp Hann window to smooth the Spo11 HpM (hits per million mapped reads) frequency, minimum length of 25 bp and 25 reads and a cut-off of 0.193 HpM. Hotspots separated by < 200 bp were merged and considered as a single hotspot. The new hotspot mask—referred to as the "Neale template"—initially identified a total of 3486 hotspots from a pooled combination of *sae2Δ ndt80Δ* and *sae2Δ ndt80Δ tel1Δ* libraries (3289 were called in *sae2Δ ndt80Δ* and 3131 in *sae2Δ ndt80Δ tel1Δ*) (**S4 Table**). 13 of the 3486 hotspots were identified at the rDNA region (position 451640–467844 kb) and therefore removed, reducing the total number of the called hotspots to 3473. From those, a total of 3195 hotspots called in this analysis were also defined by Pan et al [40] (**S5B Fig**) and 3323 by Mohibullah et al [60] using the Spo11-oligo maps performed in the Spo11-HA$_3$ and Spo11-ProtA backgrounds respectively (**S5C Fig**), thus validating that most of the hotspots positions we identify are congruent. Moreover, 278 and 150 hotspots, if compared against either Pan's template or Mohibullah's template respectively (**S5B and S5C Fig**), were exclusively defined in our *sae2Δ ndt80Δ* background strains with CC-seq technique, most of which were weak (**S5D Fig**). Similarly, 406 and 587 of the specific hotspots only defined by either Pan or Mohibullah, respectively, using the Spo11-oligo technique were also shown to be weak (**S5E and S5F Fig**). Next, to investigate whether these hotspots were Spo11-specific, hotspots were called in a *sae2Δ ndt80Δ spo11-Y135F* library identifying a total of 109 potentially false hotspots (**S5G Fig**). For this latter analysis, the cut-off was lowered to 0.125 HpM because no hotspots were called with a cut-off of 0.193 HpM. As expected, all the hotspots called in the

*sae2Δ ndt80Δ spo11-Y135F* library were weak (**S5H Fig**). Of those 109 Spo11-nonspecific hotspots, 32 were also called in our new template (**S5G Fig**), and these were all very weak. For detailed information about the mechanics of the scripts see **Hotspot_identification_V1**, **Hotspot_analysis_V1** in **https://github.com/Neale-Lab/Ndt80_LLR**. Tables of the called hotspots for each genotype are also listed here: **https://github.com/Neale-Lab/Ndt80_LLR/tree/main/HOTSPOT_TABLES_AVERAGES**.

## Script summary description

**Averaging_FullMap_tables_V1.**   This script averages individual FullMap biological replicates into a combined FullMap where the sum of HpM equals 1 million.

**Calculating background reads_V1.**   This script estimates the percentage of signal registered within the 47 largest genes—regions of presumed Spo11 inactivity—in the *S. cerevisiae* genome as an estimate of the background noise per base pair.

**Hotspot_analysis_V1.**   This script performs pairwise comparisons between datasets to study the degree of overlap, specificity, and density of the identified hotspots (generating Venn diagrams and histograms).

**Hotspot_identification_V1.**   This script identifies position and length of hotspots on single or multiple Spo11-DSB libraries. The total HpM signal is smoothed with a 201 Hann window. A cut-off of 0.193 HpM is then applied to remove the background noise. Hotspots are defined setting a minimum length of 25 bp and a minimum number of reads of 25. Hotspots separated by < 200 bp are merged and considered as a single hotspot. Hotspots are defined in each library separately and then combined to produce a single hotspot template that defines the position of every hotspot identified in the libraries.

**Hotspot_Smooth_ratios_V1.**   This script calculates and represents the hotspot fold changes between two libraries (the NormHpM ratio).

**Hotspot_table_V1.**   This script calculates the HpM and NormHpM signal included within each hotspot. Detailed description of the term heading lists is included in **Hotspot Table Definitions.docx** at **https://github.com/Neale-Lab/Ndt80_LLR**. Briefly, NormHpM refers to the total Spo11 CC-seq signal present in each hotspot (after subtraction of estimated background noise/bp) expressed as a fraction of the total signal in all the hotspot regions. Because NormHpM values utilise hotspot-specific signals (where signal density is greater), they are more robust to differences in library-to-library noise than the raw HpM values.

**NormHpM_V1.**   This script generates DSB maps representing the position and frequency of hotspots (NormHpM or NormHpChr).

**Pearson_correlation_V1.**   This script analyses the correlation between the hotspot strengths of different datasets (NormHpM and NormHpChr Pearson correlation).

**Ratio_heatmaps_V1.**   This script calculates and represents the hotspot fold changes between two libraries (NormHpM) at 50 kb bin intervals on a per chromosome base ranked by chromosome size and centred at the centromere.

**Spo11 mapping Totals_V1.**   This script represents the position and frequency of the Spo11-DSBs signal (Total HpM) along the chromosome.

## Supporting information

**S1 Fig. Calculating DSB interference. a**, Schematic representation of the potential prophase length differences between ± Tel1. In the absence of Tel1, the checkpoint may be down-regulated resulting in a reduction of the meiotic prophase length. **b–d**, Meiotic nuclear division (MI and MII) kinetics showing the individual profiles of mono- bi-, tri/tetra-nucleate DAPI-stained cells for Wild type (**b**), *sae2Δ* (**c**) and *sae2Δ tel1Δ* (**d**). Summary of bi- tri- and tetra-

previously presented in **Fig 1B**. **e**, Schematic representation of the expected effect of *ndt80Δ* mutation. Removal of *NDT80* generates cell cycle arrest in late meiotic prophase I and therefore equalizes the length of meiotic prophase regardless of the presence or absence of Tel1. **f–l**, Simplified schematics of the Southern blot method used to study DSB interference at specific loci. **f**, Diagram representing a theoretical loop domain containing two hotspots (DSB I and DSB II) that can arise independently or coincidently (double-cut, DC). **g**, Diagram representing the position of the probes and fragments that would be used to detect each of the single DSBs or the coincident double-cut by Southern blotting techniques in this theoretical scenario. The probability of both DSBs arising from independence (Expected double-cuts), can be estimated by measuring and multiplying the single DSB event frequencies. **h–j**, Three possible scenarios can result from comparing the estimated expected DC frequency with the observed DC frequency. The expected DC frequency can be higher (**h**), similar (**g**) or lower (**j**) than the observed DC frequency. **k**, The strength of interference is calculated as the negative logarithm of the observed DC frequency divided by the expected DC frequency (obtained from the product of the two individual measured DSB frequencies). **l**, Positive interference values indicate separated DSB events. Interference values close to zero suggest absence of interference, and thus, potentially, a random distribution of DSBs. Negative interference values indicate concerted DSB activity (DSB clustering). **m**, Interference as measured over time between the two hotspots within the *HIS4*::*LEU2* locus for the indicated strains (see **Fig 1** and text for further details). Error bars are standard deviation for each timepoint (n = 6 for each sample).
(TIFF)

**S2 Fig. Deletion of *NDT80* ablates short-range negative interference at the *ARE1* hotspot. A**, ***Top***, Location of *ARE1* region on chromosome III. ***Bottom***, Diagram of the *ARE1* hotspot showing Spo11-DSB positions as detected by CC-seq in hits per million (HpM; [38]), and, for Southern blotting experiments, the restriction enzyme sites, probes and size of fragments obtained from each probe. DSB interference was only measured between the main hotspot F–E and F–I. **b–c**, Representative Southern blots of genomic DNA isolated at the specified times hybridised with *TAF2* (**b**), and *PWP2* (**c**) probes. Quantified DSBs were marked in orange and not-quantified DSBs in grey. N, *NgoMIV* digested parental fragment. **d–f**, Quantification of F (**d**), E (**e**) and I (**f**) hotspots (average of 6–8 h time points). Estimation of F was corrected by adding on FI double-cuts measured with *ARE1* probe. **g–h**, As in **b–c** but with undigested gDNA samples at the indicated timepoints and hybridized with *BUD23* (**g**) and *ARE1* (**h**) probes. Quantified DCs were marked in blue and not-quantified DCs in grey. UC, Uncut parental. **i–j**, Quantification of DC signal between FE (**i**) and FI (**j**) (average of 6–8 h time points). **k–l**, Quantification of observed and expected DC frequencies between FE (**k**) and FI (**l**) using averaged data from 6–8 h time points in the indicated strains. **m–n**, DSB interference between FE (**m**) and FI (**n**) calculated for each individual repeat expressed as $-\log_2$(Observed/Expected DCs) and then averaged (see Extended methods, "Calculation of DSB interference"). In all plots, error bars indicate Standard Deviation between individual repeats (overlaid grey circles on bar graphs). For statistical analysis, a two-tailed t-test with equal variance was performed with P values indicated. n = 2 for *NDT80*[+] (from Garcia et al 2015 [56]) and n = 3 for *ndt80Δ* backgrounds.
(TIFF)

**S3 Fig. Short-range interference at the *YCR061W* hotspot. a**, ***Top***, Location of *YCR061W* region on chromosome III. ***Bottom***, Diagram of the *YCR061W* hotspot showing Spo11-DSB positions as detected by CC-seq [38] in hits per million (HpM) and, for Southern blotting experiments, the restriction enzyme sites, probes and size of fragments obtained from each probe. DSB interference was only measured between the main hotspots N–O and N–Q. **b–c**,

Representative Southern blots of genomic DNA isolated at the specified times hybridised with *YCR061W II* probe. E, *EcoRI* digested parental fragment (**b**) and P, *PstI* digested parental fragment (**c**). Quantified DSBs were marked in orange and not-quantified DSBs in grey. **d–f**, Quantification of N (**d**), O (**e**) and Q (**f**) hotspots (average of 6–8 h time points). Estimation of N was corrected by adding on NO DCs measured with the *YCR061W I* probe. **g**, As in **b–c** but with undigested gDNA samples at the indicated timepoints and hybridized with the *YCR061W I* probe. Quantified DCs were marked in blue and not-quantified DCs in grey. UC, Uncut parental. **h–i**, Quantification of DC signal between NQ (**h**) and NO (**i**) (average of 6–8 h time points). **j–k**, Quantification of observed and expected DC frequencies between NQ (**j**) and NO (**k**) using averaged data from 6–8 h time points in the indicated strains. **l–m**, DSB interference between NQ (**l**) and NO (**m**) calculated for each individual repeat expressed as–$\log_2$(Observed/Expected DCs) and then averaged (see Extended methods, "Calculation of DSB interference"). In all plots, error bars indicate Standard Deviation between individual repeats (overlaid grey circles on bar graphs). For statistical analysis, a two-tailed t-test with equal variance was performed with P values indicated. n = 2 for *NDT80*[+] (from Garcia et al 2015 [56]) and for *ndt80Δ* backgrounds.
(TIFF)

**S4 Fig. Deletion of *NDT80* does not alter Tel1 DSB interference over medium distances (*ARE1*–*YCR061W*). a**, *Top*, Location of *ARE1*–*YCR061W* region on chromosome III. *Bottom*, Diagram of the region comprised between *ARE1* and *YCR061W* hotspots showing Spo11-DSB positions as detected by CC-seq in hits per million (HpM; [38]), and, for Southern blotting experiments, the probes and size of fragments obtained from each probe. **b**, Quantification of F and N was obtained from **S2C** and **S3C Figs**, respectively. Quantification of DCs between *ARE1*–*YCR061W* was obtained from **S2H Fig**. **c**, Quantification of observed and expected DC frequencies between *ARE1*–*YCR061W* using averaged data from 6–8 h time points in the indicated strains. **d**, DSB interference between *ARE1*–*YCR061W* hotspots calculated for each individual repeat expressed as–$\log_2$(Observed/Expected DCs) and then averaged (see Extended methods, "Calculation of DSB interference"). In all plots, error bars indicate Standard Deviation between individual repeats (overlaid grey circles on bar graphs). For statistical analysis, a two-tailed t-test with equal variance samples was performed. n = 2 for *NDT80*[+] (from Garcia et al 2015 [56]) and for *ndt80Δ* backgrounds. **e**, Aggregation of interference data from all 7 loci measured in this study. The mean value of interference was plotted against the distance (in kb) between the pair of DSBs used to measure interference on a $\log_2$ scale. $R^2$, Pearson r, and *P* value of the Pearson correlation are indicated, highlighting the positive trends observed in *NDT80*[+] strains that are substantially flattened upon *NDT80* deletion.
(TIFF)

**S5 Fig. Identification of Spo11 hotspots. a**, Diagram representing the hotspot calling method (see Extended method, "Hotspot identification"). The frequency of HpM was smoothed using a 201 bp Hann window with a minimum length of 25 bp, 25 reads and a cut-off of 0.193 HpM to filter for noise signal. Hotspots separated by < 200 bp were merged and considered as a single hotspot. In this study, hotspots were identified from a pooled combination of *sae2Δ ndt80Δ* and *sae2Δ ndt80Δ tel1Δ* (Neale template). **b–c**, Venn diagrams of overlap between hotspots identified in this study by CC-seq (Neale) and hotspots identified by Spo11oligo mapping by Pan et al. 2011 [40] (**b**) or Mohibullah et al 2017 [60] (**c**). **d–f**, Distribution of hotspot frequency strengths for the total and unique hotspots identified by Neale vs Pan (**d**), Pan vs Neale (**e**) and Mohibullah vs Neale (**f**). **g**, Venn diagrams of overlap between hotspots identified in the Neale template and the non-specific hotspots identified in the *spo11-Y135F* strain. The cut-off for hotspot calling in the *sae2Δ ndt80Δ spo11-Y135F* mutant was lowered to 0.125 HpM. **h**,

as in **d–f** but *sae2Δ ndt80Δ spo11-Y135F* vs Neale template.
(TIFF)

**S6 Fig. Ndt80 genome-wide effect on a per chromosome basis.** Log$_2$ ratio of relative Spo11 hotspot intensities ±*NDT80* on all 16 chromosomes in the presence (left panel) and absence (right panel) of Tel1. Values above zero indicate a higher DSB frequency in the presence of Ndt80 and below zero a higher DSB frequency in the absence of Ndt80. Fold change was smoothed to highlight the spatial trend effect of *NDT80* deletion (black line).
(TIFF)

**S7 Fig. Tel1 genome-wide effect on a per-chromosome basis. a,** Log$_2$ ratio of relative Spo11 hotspot intensities ±*TEL1* on all 16 chromosomes in *SAE2$^+$* cells with Spo11-oligo technique (left panel) and *sae2Δ* cells with CC-seq technique in the presence (middle panel) and absence (right panel) of Ndt80. Values above zero indicate a higher DSB frequency in the presence of Tel1 and below zero a higher DSB frequency in the absence of Tel1. Fold change was smoothed to highlight the spatial trend caused by *TEL1* deletion (black line). **b,** Plot showing the Pearson correlation between ± Tel1 smoothed ratios in the presence (RATIO 1) and absence (RATIO 2) of Ndt80 for each chromosome.
(TIFF)

**S1 Table. *S. cerevisiae* strains used in this study.** All genotypes are otherwise isogenic from the SK1 strain background.
(TIFF)

**S2 Table. Oligonucleotides used in this study for Southern blots.** Oligonucleotide pairs were used in PCR to generate locus-specific probes for Southern blots. CHR indicates chromosome. PRIMERS indicate locus name and DNA primer sequence. DIGESTION indicates whether the probe was used for digested or undigested DNA (with relevant enzyme as applicable). COMMENTS indicates relevant information for this probe and/or digest combination with respect to data collection within this study.
(TIFF)

**S3 Table. Spo11-DSB Mapping libraries used in this study.** Mreads refers to million mapped Read 1 ends (the Spo11-bound CC end). For pooled data, identical genotypes were averaged with equal weighting of each library.
(TIFF)

**S4 Table. Hotspot calling statistics in various averaged libraries with thresholds used and number of hotspots present in each library and the number that overlap with the Neale CC-seq template ([60,65], this study).**
(TIFF)

**S1 Data. NDT80_Submission_Data_03.xlsx.** Data used in each graph.
(XLSX)

## Acknowledgments

We thank S. Keeney, J. Carballo and M. Lichten for sharing *S. cerevisiae* strains containing relevant constructs *(spo11-Y135F::KanMX, tel1Δ::hphNT2* and *sae2Δ::kanMX6,* respectively), and K. Caldecott and A. Oliver for sharing recombinant TDP2.

## Author Contributions

**Conceptualization:** Luz María López Ruiz, Matthew J. Neale.

**Data curation:** Luz María López Ruiz, Dominic Johnson, William H. Gittens, Rachal M. Allison, Matthew J. Neale.

**Formal analysis:** Luz María López Ruiz, William H. Gittens, Matthew J. Neale.

**Funding acquisition:** Matthew J. Neale.

**Investigation:** Luz María López Ruiz.

**Methodology:** Luz María López Ruiz, Dominic Johnson, George G. B. Brown, Matthew J. Neale.

**Project administration:** Matthew J. Neale.

**Resources:** Matthew J. Neale.

**Software:** Luz María López Ruiz, William H. Gittens, George G. B. Brown, Matthew J. Neale.

**Supervision:** Matthew J. Neale.

**Visualization:** Matthew J. Neale.

**Writing – original draft:** Luz María López Ruiz, Matthew J. Neale.

**Writing – review & editing:** Luz María López Ruiz, Rachal M. Allison, Matthew J. Neale.

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
