## [Decision Letter · Decision Letter 0]

12 Jun 2023

Dear Dr Neale,

Thank you very much for submitting your Research Article entitled 'Meiotic prophase length modulates Tel1-dependent DNA double-strand break interference' to PLOS Genetics.

The manuscript was fully evaluated at the editorial level and by independent peer reviewers. The reviewers appreciated the attention to an important problem, but raised some substantial concerns about the current manuscript. Based on the reviews, we will not be able to accept this version of the manuscript, but we would be willing to review a much-revised version. We cannot, of course, promise publication at that time.

If you decide to revise the manuscript for further consideration at PLOS Genetics, please aim to resubmit within the next 60 days, unless it will take extra time to address the concerns of the reviewers, in which case we would appreciate an expected resubmission date by email to plosgenetics@plos.org.

We are sorry that we cannot be more positive about your manuscript at this stage. Please do not hesitate to contact us if you have any concerns or questions.

Yours sincerely,

Eva H. Stukenbrock, PhD

Section Editor

PLOS Genetics

Eva Stukenbrock

Section Editor

PLOS Genetics

Reviewer's Responses to Questions

**Comments to the Authors:**

Reviewer #1: The manuscript by Ruiz et al. is a rigorous and elegant study of a type of DSB feedback control previously defined as "DSB interference", where a given DSB formation prevents other DSBs from forming nearby. In the previous study, they developed a method to measure the interference level using the ratio between the observed and expected frequency of chromatids cleaved twice (double cuts: DCs). Using this approach and the tel1∆ mutant, they demonstrated that DSB interference requires Tel1. They also discovered that the interference level varies depending on the inter-hotspot distances. Between hotspots ~20 kb apart, the TEL1 cells form fewer DCs than expected (positive interference), whereas the tel1∆ mutants form DCs equivalent to expected (no interference). In contrast, within short ranges (<15 kb), TEL1 shows no interference, while tel1∆ forms DCs more frequently than expected (negative interference, i.e., DSB clustering). To explain the negative interference in tel1∆, they proposed that these neighboring DSBs reside within one of the DSB-permissive domains subject to stochastic activation during prophase. They demonstrated by a simulation that the decrease in the frequency of the DSB-permissive domain would underestimate the interference levels, leading to negative interference.

This study by Ruiz et al. experimentally tests the above scenario. The absence of Tel1 leads to premature prophase exit that terminates global DSB formation, potentially decreasing the frequency of activating the chromosomal sub-domains. If so, eliminating the premature prophase exit in tel1∆ would equalize the number of the DSB-permissive domains with that in TEL1, thereby attenuating the negative interference. To this end, the authors used the ndt80∆ mutant, which arrests cells in prophase. Consistent with the previous hypothesis, ndt80∆ eliminated the negative interference between DSBs at short distances in tel1∆ and led TEL1 cells to exhibit positive interference. In contrast, medium-range interferences are not affected by ndt80∆. Furthermore, the authors used CC-seq, developed in a previous study, to visualize the effect of Tel1 and Ndt80 on the genome-wide distribution of DSBs and showed that chromosomal regions that preferentially form DSBs in tel1∆ correlate with the early association of proDSB factors, consistent with the expected depletion of late-forming DSBs in tel1∆ due to the premature prophase exit.

Overall, the paper is well written, the experiments are well designed, and the data broadly supports the conclusions that Tel1 mediated interference suppresses DCs within short and medium chromosomal range. Especially the experimental validation of the negative interference strengthens their fascinating proposal of stochastic activation of chromosomal sub-domains. Taken together with CC-seq maps elucidating the genome-wide impacts of Tel1 and Ndt80 on DSB distribution, this study further articulates how DSB interference and prophase length control mediated by Tel1 operate throughout meiotic prophase and shape the DSB landscape.

General comments:

I encountered difficulties in following how negative interference relates to the chromosomal sub-domains that form DSBs preferentially in the tel1∆ mutant. I thought that the premature exit from prophase would lead to a reduction in the frequency of activating sub-chromosomal domains, particularly in regions where DSBs occur later, thus further tilting the interference measurement towards negative values. In addition, I noticed that the hotspots examined in the paper are exclusively located in chrIII, where Rec114 is associated earlier. It would be beneficial if the authors could clarify the above and explain their decision to present this specific subset of hotspots, especially considering that their previous paper includes hotspots on other chromosomes.

Specific comments:

L56: ref38 uses SAE2+ and is inappropriate in this context.

L92: Homolog engagement down-regulates DSB formation in a chromosome-autonomous manner. "global" makes me feel awkward. I also suggest citing ref 90 as well.

L901-904: I wonder if the authors had explained that this correction is reasonable in the previous paper. If so, cite the paper here.

Figure 1l: I wonder if the authors could provide the numbers from replicates that were used to generate this figure panel as supplement data to help readers understand how these numbers are transformed into interference in Figure 1m.

Figure 1m: It may be worth presenting the kinetics of interference over 4-6 hours. The trend that is similar to Extended Data Figure 3i in the previous paper (Garcia et al., nature 2015) would further support the authors' model.

Figure 3a: I am curious if it is possible to calibrate CC-seq datasets to represent absolute DSB levels, for example, by quantifying the amount of the "Sonicated Spo11–DNA" or using DSB frequencies detected by Southern blotting.

Figure 3g-h: The visual correlation (e.g., stated in L449) lacks quantitative perspectives. I suggest adding scatter plots with correlation coefficients comparing panels 3g and 3h as well as others (e.g., 4f and 3h seem to correlate).

Figure 3h: The authors should explain how they converted the original data to generate this figure panel somewhere.

Figure 5: I thought the interference measurement skews towards negative because tel1∆ activates fewer domains, such as illustrated in their previous paper (Extended Data Figure 8 in Garcia et al., nature 2015). However, in Figure 5a of the current paper, TEL1+ and tel1∆ show the number of active domains. The authors might intend to convey different messages, but the current figure confused me. In addition, I found the numbers simulation presented in the previous figure helpful in following the logic. Therefore, I wonder if the authors could provide a similar simulation.

Supplementary Figure 1k: This definition of interference leads to different numerical behavior when given DSB pairs exhibit positive and negative interference (panels h and j), making it difficult to imagine the ratio between observed and expected DCs. If the authors are not strongly committed, they could change the definition of interference to straightforward ones (e.g., -log2(obs/exp)).

Supplementary Figure 1k legend: Please, fix typos.

Reviewer #2: During meiosis the introduction of DNA double-strand breaks (DSB) by Spo11 initiates homologous recombination. This process needs to be tightly regulated to ensure that DSBs form in a spatially and temporally coordinated manner throughout the genome. One key regulator of meiotic DSB formation is Tel1/ATM kinase. Two distinct roles for Tel1 that are important for the understanding of this manuscript have been characterized previously by the authors and other groups. First, Tel1 mediates DSB interference locally (across distances of 70-100 kb) through the inhibition of breaks in adjacent DSB hotspots. DSB interference influences DSB patterns and also limits overall DSB numbers. Second, Tel1 regulates DNA damage checkpoint activation and delays exit from prophase I, which is initiated by expression of the transcription factor Ndt80. How these two roles affect each other is unclear.

To address this question the authors characterize DSB formation using a series of S. cerevisiae mutant allele combinations: sae2∆, tel1∆, and ndt80∆. While the phenotype of tel1∆ in the sae2∆ background has been studied before, additional deletion of ndt80∆ now allows the authors to study the effects of Tel1 in more homogeneous populations of cells that arrest in late Prophase, thereby removing the potentially obscuring effect of premature exit from Prophase due to the lack of checkpoint activation in tel1 mutants. The authors use Southern blot assays together with genomic approaches to characterize DSB formation at different scales: overall DSB formation within a series of different hotspots, DSB interference between closely adjacent hotspots (ca. 0.7-3 kb) or across medium distances (ca. 15-30 kb) and lastly genome-wide DSB patterns.

They find that an extended Prophase due to ndt80∆ leads to a variable increase in DSB formation in the absence of Tel1, suggesting that the total DSB potential in the absence of Tel1 might have been previously underestimated, at least for some genomic loci. Interestingly, the effect of ndt80∆ on interference differs substantially between very close hotspots and across medium distances. The effect of Tel1 loss at very close hotspots (loss of interference and clustering of DSBs) is dependent on Ndt80, while at medium distance ndt80∆ has no effect. Global DSB landscapes show some remodeling in the absence of Ndt80, an effect that is stronger in the absence of Tel1. In ndt80∆ strains the increased DSBs form in regions normally less prone for DSB formation, making the overall DSB hotspot strength more homogenous across chromosomes. This is consistent with a model were shorter Prophase length (in the presence of Nd80) in tel1∆ leads to preferential DSB formation in very hot regions that previously were correlated with early and long Rec114 association time. Lastly ndt80∆ allows the authors to characterize the effect of tel1∆ genome-wide in a population of cells without premature exit from Prophase, showing that in a sae2∆ background the presence of Tel1 changes global distribution of hotspot strength across chromosomes in a largely Ndt80-independent manner.

This study addresses important, longstanding questions in the field and it provides several interesting observations and broadly useful datasets. The data are clearly presented and there are well-reasoned implications discussed for understanding how DSB patterning within individual cells plays out when measured across large populations of cells. However, there are also several significant limitations to the work in its present form that need to be addressed. The main concerns are a) there are insufficient numbers of replicates for most of the Southern blotting assays, resulting in experiments that are underpowered to detect differences rigorously; b) the analysis of the whole-genome data is somewhat superficial and limited in scope; and c) an important caveat with the use of the sae2 mutant background needs to be acknowledged.

Specific comments

1. There are a number of places where differences between genotypes are highlighted despite those differences not being statistically significant. Specifically:

- Line 246: Since the difference between sae2 tel1 and the triple mutant is not statistically significant (Fig. 2c), it is not appropriate to state that the tel1 and ndt80 effects are additive.

Lines 252-253: the ndt80 deletion did not have a statistically significant effect in either the TEL1+ or tel1 background.

- Line 261: the text says that DC frequencies are increased in the ndt80 mutant in both the presence and absence of Tel1, but the results of a statistical test are not provided in Fig. S4b (unlike in other figures).

- Line 372-374: This statement (“both TEL1 and NDT80 deletion independently increased Spo11 activity, with the greatest DSB frequency arising when both genes were deleted”) is only true for some of the measurements, but is not universally correct.

2. The above issues need to be addressed by appropriate rewriting to avoid claims based on differences that are not statistically significant. However, a bigger issue is that the measurements are intrinsically somewhat noisy, but the number of replicates was low (often just 2), so the study is underpowered to detect differences if they are there. It is essential to have more replicates to address these questions rigorously.

3. The CC-seq datasets are an important addition to the field, and several interesting observations are made with them. However, the overall analysis is some limited in scope, with only fairly limited analysis of chromosomal or chromatin features that might correlate with the observed changes. A more comprehensive analysis is warranted to determine what might be driving the variation across the genome for the effects of the tel1 and ndt80 mutations.

4. Although the use of the sae2 mutant for these studies is reasonable, using this background introduces caveats that make it challenging to extrapolate to the situation in a SAE2 background. The authors acknowledge one such caveat (line 424; and lines 471-477), having to do with the suppression of the normal DSB response in sae2 meiosis because of absence of the ssDNA that is normally generated by resection. Additionally, however, it has been established that sae2 mutants, in addition to failing to initiate DSB resection, are also constitutively hyperactivated for Tel1 activity, at least in vegetative cells (PMID: 11430828, 18245357). If this hyperactivation is also true in meiosis, then the sae2 background would be expected to artificially exacerbate the effects of a tel1 deletion compared to a SAE2 background. This issue needs to be discussed, including acknowledgment of the possibility that much of the effect of tel1 deletion seen in this study is attributable to relieving the effects of nonphysiological Tel1 hyperactivation.

5. The log-fold difference plots do not provide any visual correction for the changes in absolute DSB levels. For example, since total DSB frequencies have gone up in tel1 mutants, then the log-fold difference of the tel1 map relative to the wild-type map may give a misleading impression: regions that have experienced an increased DSB frequency but to a lesser degree than the average increase will look like their DSB frequencies have gone down instead. One problem is that the CC-seq maps are not calibrated to yield absolute DSB levels; they are only normalized relative to the sample mean (hits per million mapped reads). The caveats of using non-calibrated maps should be explained, and if possible, some indication of how to visually correct the log-fold difference maps should be provided. For example, a line or shaded area could be added to the plots to indicate an estimate for what log-fold value corresponds to no change in absolute DSB frequency; this could be estimated from the Southern blotting data (not optimal, but sufficient for the purpose here). This issue is particularly relevant to the discussion on lines 464-469, which emphasize the magnitude of the effect of Tel1: it seems quite possible that this magnitude is mostly or completely an artifact of using a sae2 background for these experiments.

6. Throughout: for bar graphs, it is becoming standard practice to superimpose individual measurements when the number of these is small, so it would be good to do this. Likewise for the interference plots (Fig. 1m and similar). Also, SEM is shown throughout, but this is rarely an appropriate choice to display experimental variation because it can be visually misleading, especially when the sample size differs between test conditions as it does here. SD should be used instead.

Minor points

1. Line 89: Mek1 is more accurately described as a paralog of Rad53, not an ortholog

2. Lines 119-124 and Figure S1a: In the introduction it is stated that Tel1 activity promotes checkpoint activation and delays Prophase exit. Figure S1a shows the same model, however the experimental data shown in Figure 1b and S1b-d only shows the rescue of Prophase exit timing in a sae2∆ background. There should be a rationalization of why repair mutants like Rad50S or sae2∆ mutants need to be used to measure the change in Prophase kinetics.

3. Line 122: The way the text is worded makes it sound like this is the first demonstration of the division delay in the sae2∆ mutant, but this has been known since the original characterization of SAE2 (McKee and Kleckner 1997). Suggest rewording and citing the earlier work

4. In Figures 1, 4 and S7 it could facilitate interpretation for the reader if the color for wild type was different. Overall, the consistent color scheme is very helpful, but it is hard to distinguish the different blue tones.

5. Line 150: not clear what the word “both” refers to (both hotspots in the region? but there are three hotspots quantified…). Please clarify.

6. Line 187: I don’t understand the rationale for averaging the 6 and 8 hr time points. As the text says, DSBs and DCs continued to accumulate from 6 to 8 hrs; since these time points are demonstrably different from one another, there does not appear to be any justification for averaging them.

7. Lines 235-237 (“over short distances Spo11 DSBs failed to display interference”) and Lines 265-267 (“Tel1-dependent DSB interference acts over both short and medium scales”) The formulation could be improved here: if I understand correctly there is interference (mechanistically) mediated through Tel1 over short and long distance, however on short distances this fails to be measured as interference because of the underlying clustering of DSBs.

8. In Figure 3 g-h the authors show the local enrichment of DSBs when Ndt80 is present in the absence of Tel1. The overall correlation is convincing but there are notable exception in chromosome 4 (and maybe also a bit in 14). Is there any particular explanation (genomic context?) why the correlation might be poorer in these regions?

9. Figure 4g the legend for the log2 fold change states -1.5 on both ends.

10. Line 317. “We further propose that it is this effect that drives the negative DSB interference (DSB clustering) that we have measured over short distances” this sentence is a bit unclear to me and requires some elaboration.

11. Line 349-352. This conclusion is difficult to interpret without the underlying manuscript.

12. General: throughout the manuscript the heterogeneity of the assayed cultures and the limitations posed by population-based assays are an important aspect of the model. I therefore think the authors should discuss in more detail how the assayed timepoints compare between the different genotypes.

13. Given the substantial quantitative variability between loci for the effects of the tel1 and ndt80 mutations when analyzed by Southern blotting, it was a bit surprising that the whole-genome data was not analyzed at these specific locations (at least, those locations present in the strains used for CC-seq).

14. Line 677: Thank you for providing the scripts. Please add a mention here that tables listing the called hotspots are also provided on the Github deposition. (If this is mentioned elsewhere, I missed it.)

Reviewer #3: In their manuscript “Meiotic prophase length modulates Tel1-dependent DNA DSB interference”, López Ruiz and colleagues analyze effects of checkpoint kinase Tel1 and prophase I exit factor Ndt80 on local meiotic DSB levels and interdependence as well as DSB global placement. Consistent with earlier findings, they report that Tel1 mediates delayed meiotic progression in a mutant that blocks DSB resection (sae2D), but also prevents the clustered formation of DSBs along the same chromatid in the same cell. Whether Tel1’s role in minimizing closely spaced DSBs is related to its role in extending the time spent at the DSB stage or whether Tel1 performs this function independent of timing issues has been unclear.

Here, the authors use the ndt80D mutation to arrest sae2 and sae2Dtel1D cells indefinitely in prophase I. They find that extended prophase partially suppresses the formation of closely spaced DSBs in absence of Tel1 while at the same time ndt80D enhances DSB formation synergistically with tel1D. The authors show that genome-wide, Tel1 is especially important for limiting DSBs in regions that load Spo11 cofactor Rec114. Enhanced DSB formation in regions occupied early by Rec114 is diminished when the accelerated prophase exit in tel1D sae2D is blocked by ndt80D, suggesting that Tel1 limits DSBs only in part via its role in delaying meiotic progression, and that non-checkpoint functions also play a role.

Overall, these findings are quite intriguing, and the data are of excellent quality. My main concerns are with the presentation of the findings.

First, there is a strange disconnect between local and global effects. The effects on DSB enhancement and DSB interference are determined by distance, but only limited attempts are made to connect these to the global effects. Do the global effects on DSB levels and placement shown in Figures 3 and 4 predict local effects at DSB sites? Could the authors point out where the DSB sites explored in Figure 1,2 , S2,3,4 are located, if possible in a main Figure? Is there a connection between hotspot-specific effects observed in Figures 1, 2 and S2-4? Can differing effects be explained by their localization along the chromosome?

Second, while struggling with Figure 3, I found myself jumping ahead to Figure 4 to find out about the effects of tel1D and tel1D ndt80D. I understand the rationale of keeping the same order of presentation from the earlier figures (sae2 vs sae2ndt80 followed by sae2 tel1 vs sae2 tel1 ndt80), but the reader is told about ndt80D effects without knowing about tel1D effects. The Abstract (L26-28) actually summarizes the findings in this order. I assume that the authors want to front load the most intriguing findings in Fig. 3g,h, but maybe there is a compromise? Would it make sense to combine into a single Figure data in Fig. 4 d,e (comparison of sae2D/tel1Dsae2D) then add the ndt80D effects (Fig. 3 d,e)? Also, can you show all effects along the same chromosome? It is frustrating that Figures 3 and 4 focus on different chromosomes (VII and IV, respectively) so it is left to the reader to put together a complete comparison of genome-wide DSB effects.

Overall, the Introduction and Results are well written, but the Discussion feels like a collection of ideas instead of putting all findings into context. Also, the discussion fails to consider alternative explanations: Could the reversibility of DSBs that carry Spo11 at their ends play a role, i.e. that in ndt80D some double cuts get resealed? Or could additional DSBs formed in ndt80D tel1D not exhibit negative interference because DSB clustering is suppressed by a factor that acts redundantly with Tel1 at later stages (NB Mec1 affects resection of later DSBs in this way, Joshi et al, Mol Cell, 2015)? Finally, a major conclusion from this work seems to be that DSB distribution and frequencies observed in sae2D/rad50S represent an underestimate (previously suggested by Borde and Lichten, Science, 2000) and that actual DSB frequencies should be derived from sae2D ndt80D? Is there any support for that conclusion from comparison with crossover/non-crossover or WT DSB maps? If you think that that is the case, you should say so and provide a revised estimate of the DSB map and frequencies. If not, then you should clearly state whether or not the DSB potential is fully exploited in WT cells. As a more general point, the authors frequently state mutant phenotypes without explaining how these phenotypes are relevant to the wild-type situation, and the manuscript would be stronger if they could clarify the relevance to the WT.

Here are some specific points. Throughout the manuscript: It is not clear why the authors use the term Spo11-DSB? It’s fine if it is supposed to indicate that DSBs in sae2D still carry Spo11 at their 5’ end. But does the reader really need a constant reminder that this is a meiosis paper and DSBs are made by Spo11?

L36: “Pairing…facilitates homologue alignment” – could you add “along the length of homologues”, without that it’s really obscure how pairing and alignment are different

L63: What are “proactive features”

L121: Why not refer to the time of 50% of max divisions so the reader knows where to look for the indicated 3 hour delay. Maybe also point out that sae2D is a mixed delay/arrest whereas the fraction of arrested cells is much smaller in absence of Tel1.

L133: Please define in the introduction what you mean by “DSB (forming/formation) potential” and use one term consistently throughout the paper. The idea that (during wild-type meiosis ?) some DSBs have potential but do not fulfill that potential is really not that obvious. The term currently appears in the Abstract and then again in the Results section.

L150: Not sure what “both” refers to

L195: “even though Tel1… DCs inhibited” is redundant

L241: Could that be: “Tel1 controls DSB interference over medium distances independent of Ndt80”?

L269: Again, could you rephrase that subheading to indicate how the mutant phenotype informs us about the WT, rather stating the mutant phenotype?

L370: Sae2 is referred to as an Mre11 activator, but ref 29 (Keeney, Kleckner) seems to be incorrect, and the other two papers (McKee; Prinz) do not speak to the mechanism by which Sae2 affects resection. Ref 25 (Cannavo and Cejka. 2014) might be more appropriate if the authors want to refer to Sae2’s activity.

L372: If tel1D and ndt80D increased Spo11 activity independently, as stated, then both single mutants should result in increased DSBs. But that is not the case, for sure not in Fig. 1f-h. It looks more like a synergistic effect, where ndt80D affects DSB levels only in a tel1D background (Fig. S3f is the exception where ndt80D does increase the levels of DSBs by itself). The conclusion should be that Tel1 becomes even more critical in limiting DSBs in an ndt80D background.

L390: The sentence about detection of positive interference comes out of nowhere and needs to be embedded in the argument.

L395: “these effects”: It’s unclear what effects you mean and how this is related to the previous paragraphs

L400: “If the activation step that limits total DSB potential…” I don’t understand this sentence. Isn’t it evident from the fact that different cells form DSBs in different places that “active domains vary in their chromosomal location across the cell population”? The real question seems to be whether all cells have the same distribution of DSB potential along their genomes, and only the activation is different in each cell, or the DSB potential already is different in different cells. The discussion should explain how findings in this paper contribute to answering this question, specifically: what are the contributions from sufficient time spent in prophase (mediated by Ndt80) versus other regulatory functions (mediated by Tel1).

L414: Please explain why you think that eliminating NDT80 makes the population more homogeneous? The sae2 tel1 population seems to be as homogeneous as the WT – wouldn’t non-homogeneous mean that a subpopulation of cells arrests or the population is biphasic? If you mean that all potential DSB sites get an opportunity to become activated due to the extended duration of prophase I, then that should be stated clearly.

L419: I thought that this is one of the questions that this paper tries to answer: whether Tel1 plays a dual role in DSB interference and checkpoint activation? I think that point would deserve some elaboration.

L466: Please clarify this section. I’m assuming this should say: tel1D results in increased DSB formation in certain chromosome regions independent of the presence or absence of Ndt80?

Figure 1b: Please show here or in the Supplement that you have confirmed that all mutants arrest in the ndt80D background.

Figures 3f-g and 4f-g: Can you clearly mark the chromosome ends? Or make the background black? The surrounding color is very similar to the shading of 0-fold changes.

Figure 5: In 5b, ndt80D should be written in lower case letters. I feel the loops in cells 1-4 in the upper half of the figure could be easily be omitted or only shown once, providing more space for the all important lower half of the figure.

Legend Figure 5: It should say upfront in the Figure that this model refers to short-distance interference. The phrase “lower-than-expected calculations of DSB interference” is a real brain twister: (Positive) interference refers to instances where the observed outcome is lower than the (calculated) expected. Aren’t you simply referring to underestimates? Also, how can the DSB frequency be underestimated when DSBs are a measured entity? It seems the calculations only work out if there are 2 clustered DSBs in Cell 3, but it is unclear why Cell 3 in the tel1D ndt80D mutant receives two cuts while Cell 1 receives 3?

**Have all data underlying the figures and results presented in the manuscript been provided?**

Reviewer #1: Yes

Reviewer #2: Yes

Reviewer #3: Yes

PLOS authors have the option to publish the peer review history of their article (what does this mean?). If published, this will include your full peer review and any attached files.

Reviewer #1: No

Reviewer #2: No

Reviewer #3: No

---

## [Decision Letter · Decision Letter 1]

17 Oct 2023

Dear Matt,

Thank you very much for submitting your Research Article entitled 'Meiotic prophase length modulates Tel1-dependent DNA double-strand break interference' to PLOS Genetics.

The manuscript was fully evaluated at the editorial level and by independent peer reviewers. While two of the reviewers were satisfied with the revision, Reviewer #2 was not, and I regret to say that I agree completely with their concerns. Based on this, we will not be able to accept this version of the manuscript, but we would be willing to review a much-revised version. We cannot, of course, promise publication at that time.

Reviewer 2 points out that several of the comparisons made in the manuscript, in particular those centered around Figure 2, are not statistically significant. This being the case, no further conclusions about relationships can be made. That's is, bottom line. As stated by the reviewer, in such cases all that can be said is that one set of values is not significantly different from the other; this negates further conclusions based on differences but also further conclusions based on similarity. I strongly suggest that the manuscript be revised as indicated by this reviewer, including the "Additional points" that refer to two other concerns regarding the manuscript.

In addition, now that the individual data points are available, it is clear that there are substantial technical problems, in particular with DSB frequencies in *sae2∆ ndt80∆ tel1∆* strains from *FRM2*-probed blots, where only two replicates are present and the difference in values between the two is greater than three-fold. This issue was pointed out in previous reviews, and there certainly has been sufficient time between submissions to address it. While I am not going to insist that additional replicates be done (perhaps technical replicates by reprobing filters with the "other" probe?), you must know that the data in their current form really do compromise credibility. Is there any way that you can begin to address the source of this variability? For example, in Figure 1, do frequencies of DSB II from MXR2-probed gels agree with those on HIS4-probed gels? In Figure 2, do frequencies of DSBs at leu2::hisG on CHA1-probed gels agree with those on FRM2-probed gels?

At any rate, as a minor point, current convention is that SD should not be used for error bars when only two values are present--in such circumstances, error bars should report range.

Finally, unless I missed a file, the data underlying each graph has not yet been reported in an accessible way. Data underlying each figure panel should be reported in a separate table that can be readily identified as belonging to that figure panel. If a point on a graph represents the mean of two or more values, then the table should contain those individual values as well as the mean. The current Excel file is impenetrable, and much of the data reported in the paper are missing from this file (c.f. panel 1b, 1f, 1g, etc.).

If you decide to revise the manuscript for further consideration at PLOS Genetics, please aim to resubmit within the next 60 days, unless it will take extra time to address the concerns of the reviewers, in which case we would appreciate an expected resubmission date by email to plosgenetics@plos.org.

We are sorry that we cannot be more positive about your manuscript at this stage. Please do not hesitate to contact us if you have any concerns or questions.

Yours sincerely,

Michael Lichten, Ph.D.

Academic Editor

PLOS Genetics

Eva Stukenbrock

Section Editor

PLOS Genetics

Reviewer's Responses to Questions

**Comments to the Authors:**

Reviewer #1: The authors responded satisfactorily to all my comments. I recommend accepting the revised manuscript. I appreciate the authors' efforts to conduct technically challenging experiments and draw robust conclusions successfully. Here are my final minor comments

I wonder if plotting all DSB interference measurements as a function of the distance between hotspots could further emphasize the robustness of the authors' conclusions.

How about activating one more domain on the left in Figure 5a to match the text below "Probability of domain activation = 0.5"?

Reviewer #2: The authors have addressed most of the concerns raised in the previous reviews. However, there remain concerns about conclusions drawn where differences are not statistically significant. As noted in prior review comments, the study is underpowered to make some of the conclusions the authors wish to draw. The authors have argued out of providing the additional replicates needed to resolve these issues, which is fine, but they must accept that they can’t make claims that do not have statistical support. Note that switching post hoc to using one-sided p values (as mentioned in the response to reviews) would not be appropriate, and cannot be used to justify making claims that lack statistical support.

Examples:

Lines 259-261: the difference between sae2 ndt80 tel1 and sae2 tel1 is not statistically significant. Therefore, no conclusion can be drawn here, and the data most definitely do not suggest an additive relationship, as asserted. The observation can be described, but the claim that the data suggest the additive relationship needs to be deleted. The fact that DSBs at another hotspot (leu2::hisG) do not show any similar trend further undermines confidence in the point the authors wish to make.

Line 267: again, not statistically significant (p = 0.12). Looking at the data here (Fig 1f), two replicates showed an increase and one replicate showed a decrease relative to the mean for sae2 tel1. There is simply not enough information here to draw a conclusion. Throwing in “albeit variable” to qualify the claim of an increase is not appropriate. The claim needs to be deleted.

Line 268: this doubles down on the claims above about there being increases in both the single cut and double cut frequencies. The statement here is simply not supported by the data. There really isn’t any need to make this claim anyway, so it can be safely deleted.

Lines 275-281: The data provide no evidence to support the conclusion that DC frequencies are “modestly increased” by ndt80 deletion in the presence or absence of TEL1, as claimed. This needs to be deleted. The conclusions about interference are mostly fine, except that the the “weakening” when comparing sae2 and sae2 ndt80 is again not significant.

Additional points:

Lines 249 and 285: I find it confusing to couch the conclusions here in terms of “underestimating” interference. If I am following the argument, the point is that the numerical value for the interference calculation is lower (more negative) in the presence of NDT80 for short distances, and gets larger (closer to zero) in the absence of NDT80. If so, then say that. The problem with saying that the interference is “underestimated” is that a more negative value actually means STRONGER (negative) interference than a value close to zero. Interference can be either positive or negative, and the further you are from zero, the stronger each type is. It is not really until Fig 5 (Discussion) that this idea of underestimation of interference is properly explained. I’d suggest using different wording in the earlier sections of Results.

Lines 390-395 and 533-539: It’s not appropriate to make assertions like this based on unpublished data. Whether this information has been presented at conferences (as stated in the response to reviews) is immaterial. These sections need to be deleted and this paper constrained to discuss what is actually presented here: the authors can make these other points in the other paper when it comes out.

Line 611: GEO accession is still described as pending

Reviewer #3: The authors have made revisions that appropriately address reviewer comments.

**Have all data underlying the figures and results presented in the manuscript been provided?**

Reviewer #1: Yes

Reviewer #2: **No: **CC-seq data need to be deposited. The manuscript still describes this as pending (line 611)

Reviewer #3: Yes

PLOS authors have the option to publish the peer review history of their article (what does this mean?). If published, this will include your full peer review and any attached files.

Reviewer #1: No

Reviewer #2: No

Reviewer #3: No

---

## [Decision Letter · Decision Letter 2]

17 Jan 2024

Dear Matt,,

We are pleased to inform you that your manuscript entitled "Meiotic prophase length modulates Tel1-dependent DNA double-strand break interference" has been editorially accepted for publication in PLOS Genetics. Congratulations!

Yours sincerely,

Michael Lichten, Ph.D.

Academic Editor

PLOS Genetics

Eva Stukenbrock

Section Editor

PLOS Genetics

Comments from the reviewers (if applicable):

Reviewer's Responses to Questions

**Comments to the Authors:**

Reviewer #2: The authors have responded well to prior critiques. The justification for omitting the outlier blot(s) appears to be quite reasonable, and I appreciate the transparency from the authors in providing this explanation.

**Have all data underlying the figures and results presented in the manuscript been provided?**

Reviewer #2: Yes

PLOS authors have the option to publish the peer review history of their article (what does this mean?). If published, this will include your full peer review and any attached files.

Reviewer #2: No

**Data Deposition**

http://datadryad.org/submit?journalID=pgenetics&manu=PGENETICS-D-23-00486R2

**Press Queries**

---

## [Editor Report · Acceptance letter]

26 Feb 2024

PGENETICS-D-23-00486R2 

Meiotic prophase length modulates Tel1-dependent DNA double-strand break interference 

Dear Dr Neale, 

We are pleased to inform you that your manuscript entitled "Meiotic prophase length modulates Tel1-dependent DNA double-strand break interference" has been formally accepted for publication in PLOS Genetics! Your manuscript is now with our production department and you will be notified of the publication date in due course.

With kind regards,

Anita Estes

PLOS Genetics

On behalf of:
